# Diurnal variation of amplified canopy urban heat island during heat wave periods in the Beijing megacity: Roles of mountain-valley breeze and urban morphology

Tao Shi[1], Yuanjian Yang[2*], Ping Qi[1], Simone Lolli[3]

[1]School of Mathematics and Computer Science, Tongling University, Tongling, 244000, China
[2]School of Atmospheric Physics, Nanjing University of Information Science and Technology, Nanjing, 210044, China
[3]CNR-IMAA, Contrada S. Loja, 85050 Tito Scalo (PZ), Italy

*Correspondence to*: Prof. Yuanjian Yang (yyj1985@nuist.edu.cn)

**Abstract.** Under the background of global warming and rapid urbanization, heat waves (HW) have become increasingly prevalent, amplifying the canopy urban heat island intensity (CUHII). Beijing megacity, characterized by rapid urbanization, frequent high-temperature events, and exceptionally complex terrain, presents a unique case to study the synergies between HW and CUHI. However, research exploring the formation mechanisms of the amplified CUHII during HW periods (ΔCUHII) in the Beijing megacity from the perspectives of mountain-valley breeze and urban morphology remains scarce. This study found that compared to non-heat wave (NHW) periods, the average daily CUHII during HW periods significantly increased by 59.33%. On the urban scale, the wind direction reversal of the mountain-valley breeze might contribute to the north-south asymmetry in the ΔCUHII. On the street scale, wind speed was inversely proportional to the ΔCUHII. In addition, the ΔCUHII was closely related to urban morphology, particularly the three-dimensional indicators of buildings. During the mountain breeze phase, high rise with lower sky view factors (SVF) exhibited a more pronounced effect on amplifying CUHII compared to low rise with higher SVF. Conversely, during the valley breeze phase, high rise exerted a dual influence on amplifying CUHII. Our findings provided scientific insights into the driving mechanisms of urban overheating and contributed to mitigating the escalating risks associated with urban excess warming.

## 1 Introduction

The interaction between climate and urbanization has become one of the key topics in current global climate change research (Seto et al., 2012; Ding, 2018), e.g, the interaction between increased HW events and enhanced CUHII (Li & Bou-Zeid, 2013; Founda et al., 2015; Khan et al., 2020; Ngarambe et al., 2020; Zinzi et al., 2020). Even during the hiatus of global warming, the frequency and duration of HW events also exhibited an increasing trend worldwide, posing significant challenges to the urban thermal environment management and public health safety (IPCC,2023; Patz et al., 2005; Xu et al., 2016; Yang et al., 2017). With the acceleration of urbanization and population aggregation, the CUHI in megacities has become increasingly prominent (Liu et al., 2007; Zheng et al., 2018a; Yang et al., 2020), exacerbating the occurrence of regional extreme high-temperature events (Zong et al., 2021), and seriously affecting urban development and the health of

residents (Gao et al., 2015; Jiang et al., 2019). For instance, compared to NHW periods, the average CUHII in Shanghai has increased by 128.91% during HW periods (Yang et al., 2023), while the maximum CUHII in Seoul has increased by 4.5℃ during HW periods (Ngarambe et al., 2020). The rate of contribution of urbanization to the excessive mortality caused by high temperatures can reach more than 45% in high-density urban areas (Zong et al., 2022). Therefore, in the context of global warming and rapid urbanization, it is very important to explore various driving mechanisms for the synergies between HW and CUHI.

In terms of natural impact factors, the uneven temporal and spatial distribution of urban excess warming is significantly affected by local circulation in different geographical environments (Zhang et al., 2011; Zhou et al., 2020; Chen et al., 2022). A few studies focused on the impact of local circulations on the ΔCUHII (Yang et al., 2023; Xue et al., 2023). Mountain-valley breeze represents a local circulation within mountainous terrains induced by the mesoscale-to-small-scale thermal effects between mountain and valley. In detail, the air in valleys and slopes warms up more significantly than the free atmosphere at the same altitude in mountainous regions during the daytime, leading to a temperature gradient that drives the air to ascend along the slopes, forming the valley breeze. In contrast, the adjacent air rapidly cools and becomes denser in mountainous regions during nighttime as the radiative cooling over the underlying surface, thereby flowing downslope, giving rise to mountain breeze (Jiang et al., 1994; Fu, 1997; Dong et al., 2017). The characteristics of the mountain-valley breeze are contingent upon various factors, including local topography and large-scale synoptic conditions (Whiteman et al., 1993; Zängl, 2009), atmospheric stability (Rao & Snodgrass, 1981; Whiteman & Zhong, 2008), underlying surface types (Wang et al., 2015; Letcher & Minder, 2018), and insolation conditions (An et al., 2002). Notably, under the effects of the urban underlying surface surrounding mountains, the CUHI circulation and mountain-valley breeze at mountain slopes interact and reinforce each other (Li et al., 2017). However, a limited number of studies have delved into the influence of mountain-valley breeze on the synergies between HW and CUHI (Xue et al., 2023; Yang et al., 2024). During HW periods, the mountain-valley breeze enhanced the vertical turbulent heat transfer, and improved ventilation conditions reduced aerosol concentration (the urban canopy received more short-wave radiation), both beneficial to the amplifying CUHII in Lanzhou (Xue et al., 2023). The current understanding of how local circulations modulate the ΔCUHII is still in the exploratory stage.

From the perspective of anthropogenic impact factors, urban morphology is also an important factor influencing the local thermal environment (Oke, 2006; Merckx et al., 2018; Tian et al., 2019). Building height has a complex impact on solar radiation during daytime and long-wave radiation at night (Srivanit & Kazunori, 2011; Oke et al., 2017), while building density alters the wind field in open spaces (Erell et al., 2011; Ao et al., 2019). Local climate zones (LCZs) have defined the range of values for parameters such as land cover, average building height, and sky view factor (SVF) within a climate zone, enabling the discovery of the characteristics of thermal environmental variations within cities (Stewart & Oke, 2012; 2014). Scholars have studied the urban excess warming in different LCZs, advancing the quantitative research on the synergies between HW and CUHI (Ngarambe et al., 2020; Zheng et al., 2022; Xue et al., 2023; Yang et al., 2023). The intensity, frequency, and duration of HW events in LCZ1 and LCZ2 (dominated by dense mid-rise and high-rise, respectively), are

significantly stronger than in other types of climate zones (Yang et al., 2023). LCZs are a comprehensive indicator of urban morphology, and the aforementioned studies have not quantified the contribution of different urban morphological parameters to the local thermal environment, nor have they taken into account the nonlinear driving effects of urban morphology on the local thermal environment (Alonso & Renard, 2020; Chen et al., 2022).

In the context of the frequent HW events, rapid urbanization has induced a pronounced CUHI effect in the Beijing megacity.
Coupled with Beijing's exceptionally complex terrain, the megacity presents a unique case study for investigating the synergies between HW and CUHI. However, previous research has predominantly centered on the spatiotemporal variations of the ΔCUHII in the Beijing megacity (Zong et al., 2021; Jiang et al., 2019), leaving a gap in knowledge regarding the driving mechanisms of local circulation and urban morphology on amplifying CUHII during HW periods. To address this, this study focused on the Beijing megacity, utilizing automatic weather station observations, remote sensing data, the LCZ
dataset, and machine learning models. We conduct a thorough analysis of the spatiotemporal characteristics and forming mechanisms of the ΔCUHII. Ultimately, our objective is to strengthen technical support for high-temperature forecasting, improve human settlements, and inform urban planning and management strategies.

## 2 Data and methodology

### 2.1 Study Area

In 2022, Beijing's population had exceeded 20 million and the built area was more than 1,400 km$^2$, making it one of the most urbanized cities in China. The terrain of Beijing is exceptionally complex, northerly bounded by Yan Mountains and Taihang Mountains in the west. The altitudes of those mountains exceed 2,000 m. The northeastern region comprises hilly terrain, while the southern region is dominated by plains. The area extending from the east to the southeast is a zone where land and sea intersect, bordering Bohai Bay. Under the control of a weak weather system with no clouds or few clouds (You
et al., 2006; Liu et al., 2009; Dong et al., 2017), the mountain-valley breeze formed by the complex terrain plays a dominant role in the atmospheric circulation of the Beijing area (Liu et al., 2009; Miao et al., 2013; Dou et al., 2014). The near-surface boundary layer features including wind and temperature fields during summer in Beijing, China are investigated by numerical simulation (Hu et al., 2005). The results revealed a notable CUHI effect in the city center, with the boundary layer wind field being significantly influenced by the mountainous terrain in the northwest. Furthermore, the impact of
mountainous terrain on the lower atmospheric boundary layer in the Beijing area during summer was investigated (Cai et al., 2002; You et al., 2006). They discovered that the influence of mountain-valley winds could extend to cover the plain regions around Beijing to a significant degree.

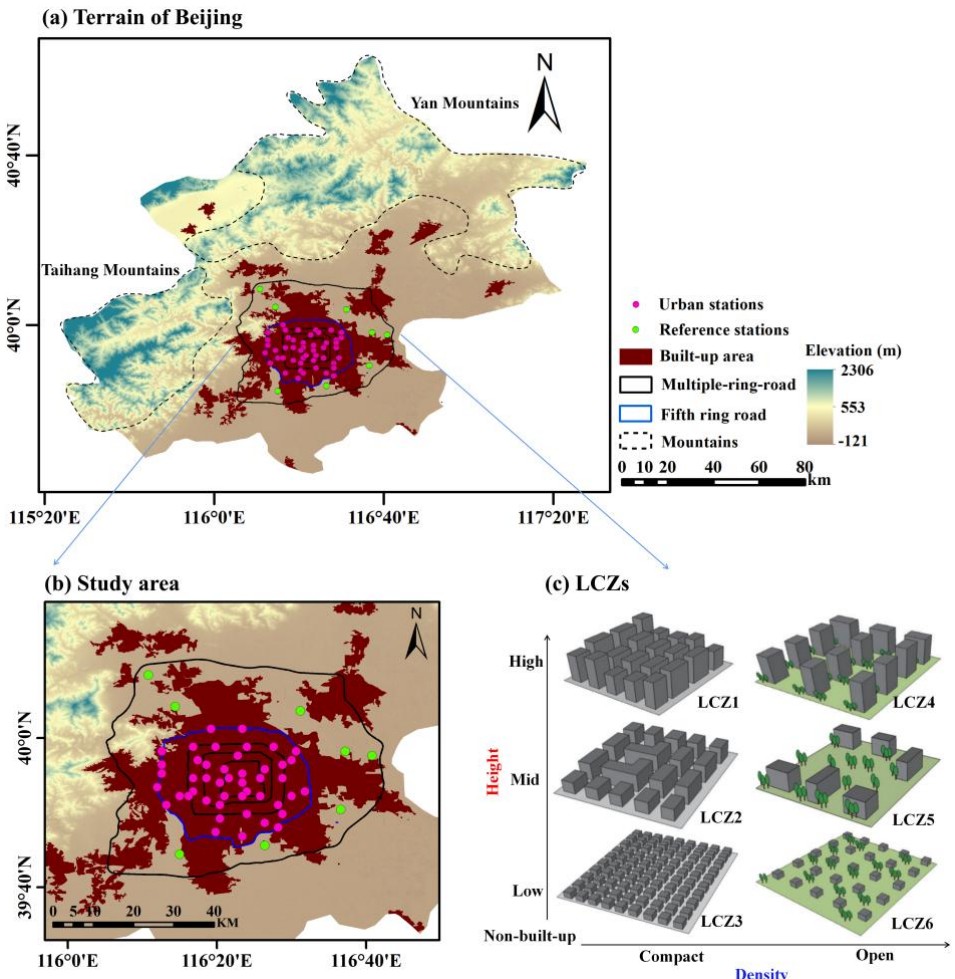

**Figure 1: Overview of the study area. (a) Terrain and land use of Beijing. (b) Distribution of urban stations and reference stations in the built-up area of Beijing. (c) Empirical examples of the typical LCZ types.**

## 2.2 Data

### 2.2.1 Urban morphology datasets

Land cover modulates the energy exchange, water, and carbon cycle between different regions of the Earth. In the past few decades, the land cover in China has greatly changed with the development of the economy. The annual China Land Cover Dataset (CLCD) is a dynamic data set accounting for land use in China released by Yang & Huang (2021). They made the land cover datasets with a spatial resolution of 30 m based on 335,709 Landsat images on Google Earth Engine. The latest datasets contain information on China's land cover from 1985 to 2021, and the overall precision of land classification is 80%. The LCZ datasets in this article were provided by the Institute of Urban Meteorology, China Meteorological Administration.

Stewart & Oke (2012) introduced the concept of LCZs, defining them as geographical regions spanning from hundreds to thousands of meters in size. These zones are characterized by uniformity in land use patterns, comparable spatial arrangements and building materials, and congruent patterns of human activity. The building skyline and floor data of the electronic map were extracted from Gaode Maps using Python language. The height of each floor was set to be 3 m, to obtain information on the height of the buildings within the study area.

**2.2.2 AWS observation data**

The hourly AWS observation data used in this article were obtained from the China Meteorological Data Service Center (http://data.cma.cn/en), which primarily includes near-surface air temperature, wind speed, wind direction, humidity, precipitation, etc. To ensure the rigor of the data, we conducted quality control on the observed meteorological data at ground stations. Following previous methods (Yang et al., 2011; Xu et al., 2013), missing values in observation sequences

were replaced with the average of synchronous observation data from the five nearest stations surrounding the given station, and stations with excessive error records were excluded. Consequently, AWS observation data were used for a detailed analysis of the spatio-temporal characteristics of the near-surface thermodynamic field in Beijing.

**2.3 Methods**

**2.3.1 Calculations of  CUHII and definition of HW**

In general, scholars define CUHII as the temperature difference between the urban station and the reference station (Ren et al., 2007; Shi et al., 2015). The Fifth Ring Road in Beijing, with a length of 98.6 km and an area of approximately 300 km$^2$ (depicted by the blue loop in Fig. 1), essentially covers the primary regions of the built-up area (Yang et al., 2013). Therefore, in this study, we have designated stations within the Fifth Ring Road as urban stations. The selection of reference stations is

crucial for calculating the CUHII. In this study, we first identified reference stations with significantly lower temperatures than those of urban stations. Additionally, the reference stations must be located more than 50 km away from the city center, in a rural environment, predominantly situated within areas of sparse trees and shrubs (Yang et al., 2023). They should also be evenly distributed across different directions of the entire city. According to these criteria, eight reference stations were selected (green plot in Fig. 1), with an average altitude of 39.6 m, which is only 8.8 m lower than the average altitude of 45

urban stations (red plot in Fig. 1). The summer CUHII of urban stations could be obtained by calculating the air temperature difference between the urban stations and the reference stations during the summertime.

Due to variations in climatic backgrounds, geographical conditions, socioeconomic factors, and other variables, different standards have been adopted for studying HW events across the world. The World Meteorological Organization suggests that an HW event occurs when the daily maximum temperature exceeds 32°C and persists for more than three consecutive

days. The National Oceanic and Atmospheric Administration of the United States defines an HW index that combines temperature and relative humidity, issuing a heat alert when the HW index exceeds 40.5°C for at least 3 hours in two consecutive days during the daytime, or when it is forecasted to exceed 46.5°C at any time. The Royal Netherlands Meteorological Institute stipulates that an HW event occurs when the daily maximum temperature is above 25°C for more

than five consecutive days, with at least three of those days having a maximum temperature exceeding 30°C. In contrast, the China Meteorological Administration (CMA) defines an HW event as a period when the daily maximum temperature exceeds 35°C for three consecutive days. In this study, the HW criteria published by the CMA were finally adopted. Considering that the daily maximum temperature at urban stations can be influenced by urbanization, this study utilizes the daily maximum temperature from reference stations to identify HW events. During the summer, if more than two reference stations experience an HW event on a given day, the day during the HW event is defined as an HW day; otherwise, it is considered an NHW day. In addition, the ΔCUHII was obtained by subtracting the summer CUHII during the NHW periods from the summer CUHII during the HW periods, providing valuable insights into the impact of extreme heat events on the environmental parameter in question. (Yang et al., 2023; Xue et al., 2023).

### 2.3.2 Calculation of mountain-valley breeze

In the Beijing region, the most significant local circulation is the mountain-valley breeze. During the day, the wind blows from the valley to the mountain due to the thermal difference between the valley and its surrounding air, while at night, the wind reverses direction, blowing from the mountain to the valley (Tian & Miao, 2019). However, local circulation can be difficult to observe as a result of the influence of mesoscale weather patterns. Therefore, when analyzing mountain-valley breeze, it is crucial to remove the effects of mesoscale wind field. Referencing relevant methods (Cao et al., 2015; Zheng et al., 2018), the mountain-valley breeze is extracted and the details are shown below. Firstly, the hourly wind data from each observation station were decomposed into the components of u (east-west direction) and v (north-south direction). From June to August between 2016 and 2020, the average values of the hourly wind components were calculated, yielding hourly average values $\bar{u}$ and $\bar{v}$. Subsequently, the diurnal average values U and V were obtained by averaging all the hourly average values $\bar{u}$ and $\bar{v}$, respectively. The hourly anomalies u' and v' were then derived by subtracting the diurnal average values U and V from the hourly average values $\bar{u}$ and $\bar{v}$, respectively. The diurnal average values U and V can be interpreted as the systematic wind or background wind, while the hourly average values $\bar{u}$ and $\bar{v}$ can be considered as the actual wind. The local wind u' and v' obtained by subtracting the systematic wind from the actual wind, can be utilized in studies focused on regional local circulations, in particular for the mountain-valley breeze.

### 2.3.3 Indicators of urban morphology

Numerous two-dimensional (2D) indicators and three-dimensional (3D) indicators have been used to quantify urban morphology (Zakšek et al., 2011; Tompalski & Wężyk, 2012; Berger et al. 2017). Here, we selected six 2D indicators and six 2D indicators to measure the morphological characteristics of buildings within a 500 m buffer zone surrounding the AWS (Oke, 2004), as shown in Tab. 1. Horizontal indicators represent the physical properties of the underlying surface and were used to explore the effect of the underlying surface on the air temperature. Vertical indicators reflect the complex effect of landscape patterns on wind fields and solar radiation within neighborhoods. The calculation of horizontal and vertical urban morphology indicators was based on land cover datasets and building height information.

**Table 1: The 2D and 3D urban morphology indicators involved in this paper.**

| Indicators | Description |
| --- | --- |
| **2D** | |
| BCR | Building cover ratio represents the proportion of the roof of the buildings to that of the buffer zone. |
| NEAR | Mean distance between adjacent buildings. A lower value of this metric indicates a higher density of buildings. |
| NP | Number of buildings patches indicates the degree of fragmentation of buildings within a given area. |
| SPLIT | Splitting index represents the degree of separation of landscape segmentation. The greater the value, the more fragmented the landscape. |
| AI | Aggregation index, which represents the connectivity between patches of each type of landscape. The smaller the value, the more discrete the landscape. |
| L/W | Length-width ratio of buildings is a metric that represents the shape characteristics of buildings. |
| **3D** | |
| H | The height of buildings represents the average height of all buildings in the buffer zone. |
| H-max | Maximum height of buildings in the buffer zone. |
| H-std | The standard deviation of building height in the buffer zone. |
| FAR | Floor area ratio represents the ratio of the sum of gross floor area to total buffer zone. The higher the FAR, the greater the amount of building floor area per unit of land area. |
| CI | Cubic index represents the ratio of the building volume to the total study volume. It indicates a higher degree of built-up density or spatial occupation within the buffer zone when the value is larger. |
| SVF | Sky view factor represents the ratio of radiation received by a planar surface from the sky to that received from the entire hemispheric radiating environment. It ranges from 0 to 1, with 0 indicating complete obstruction and 1 indicating complete exposure. |

175

### 2.3.4 Fitting model

Multiple linear regression analysis is a statistical method to determine the quantitative relationship between dependent variables and multiple independent variables (Li, 2020). Although the traditional linear regression model is straightforward and intuitive, it frequently falls short in effectively addressing intricate non-linear relationships. Support Vector Regression (SVR) is widely used as an effective supervised learning method. By introducing the concept of support vectors, SVR improves the fitting ability of data while maintaining the complexity of the model (Smola & Schölkopf, 2004). The Random Forest (RF) model, a popular and highly flexible machine learning approach (Breiman, 2001), can simulate complex nonlinear relationships between predictive values and diverse predictors (Hastie et al., 2009). The RF model exhibits low sensitivity to outliers and missing values in data sets, and due to the law of large numbers, it is less prone to overfitting. Previous studies have shown that the RF model is effective in fitting complex problems and measuring the importance of factors (Tan et al., 2017; Yu et al., 2020).

Taking the ΔCUHII as the dependent variable, the influencing factors were input into the linear model, the SVR model, and the RF model including 2D indicators and 3D indicators as independent variables. The evaluation of the influence of urban morphology on the ΔCUHII was conducted by assessing the significance and importance scores of the input parameters employed in the model. The construction of various models, the importance scores of the influencing factors, and the significance testing were implemented using Python code.

### 3 Results

### 3.1 The spatial-temporal pattern of urban excess warming

In the context of climate warming, the vast urban expansion has led to a constant increase in urban population density, while human activities have generated significant anthropogenic heat and pollutant emissions, thereby amplifying urban excess warming to a certain extent.

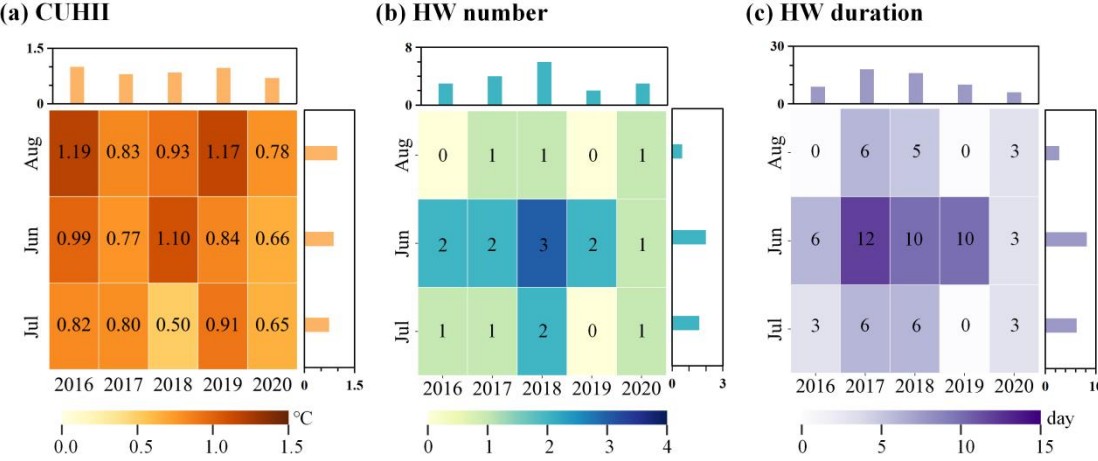

**Figure: 2 The temporal variations of CUHII and HW events from 2016 to 2020. (a) the CUHII based on urban stations and reference stations, (b) number of HW events based on reference stations, and (c) duration of HW events based on reference stations.**

Fig. 2 illustrates significant inter-annual variations in CUHII, HW number and HW duration in Beijing. The most prominent years for urban excess warming, specifically in terms of CUHII, were 2016 and 2019, with intensities of 1.00°C and 0.97°C respectively. In these two years, the HW numbers were 3 times and 2 times, while the HW duration were 9 days and 10 days respectively. The occurrence and persistence of such widespread high-temperature events in the North China region are closely related to specific atmospheric circulation anomalies. Potential influencing factors include the circulation pattern of the 500 hPa geopotential height field (Sun et al., 2011), ocean-atmosphere anomalies such as changes in the cold and warm phases in the equatorial central and eastern Pacific, as well as the position and intensity of the warm high-pressure ridge over the continent or the subtropical high over the northwest Pacific (Wei & Sun, 2007). Furthermore, there are distinct intraseasonal variations in CUHII and HW events in Beijing. HW events are stronger in June and July, averaging 6.2 days per month, significantly higher than in August. Intraseasonal variations in urban excess warming may be associated with combined differences in weather conditions, including precipitation, wind vectors, cloud cover, fog, and air pollution (Unger et al., 2001; He BJ, 2018; Chen et al., 2022).

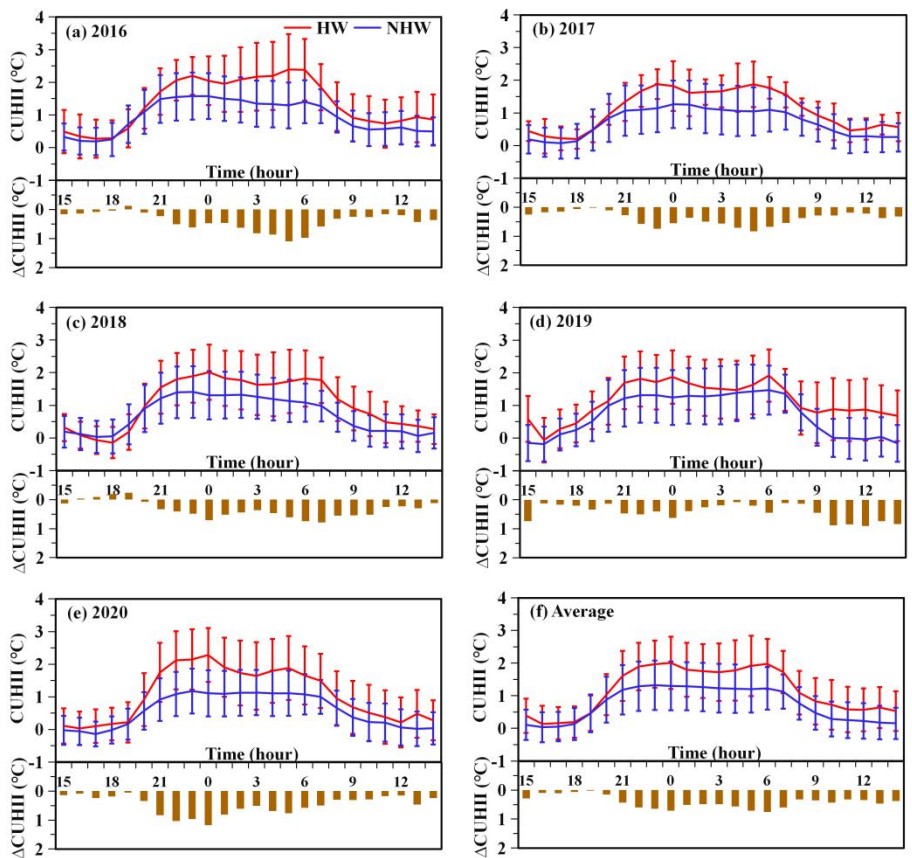

**Figure: 3 Summer diurnal variation (Beijing time, BJT) and standard deviation of CUHII during HW periods and NHW periods based on the urban stations and reference stations in Beijing. (a) 2016, (b) 2017, (c) 2018, (d) 2019, (e) 2020, (f) average value from 2016 to 2020. The red line represents the CUHII during HW periods, while the blue line depicts the CUHII during NHW periods. The bars indicate the ΔCUHII during HW periods.**

In Fig. 3, the summer diurnal variation of the CUHII in Beijing shows a U-shaped fluctuation. CUHII started to slowly decrease from 06:00 Beijing Time (BJT) and hit its lowest point at 16:00 BJT. Then, CUHII gradually increased and remained at a high plateau consistently from 22:00 until 05:00 the next day. The diurnal variation of the ΔCUHII was also examined in this study. Apart from 19:00 in 2016 (Fig. 3a) and 2018 (Fig. 3c), the hourly ΔCUHII values for all other years were positive. Taking the annual average as an example (Fig. 3f), during HW periods, the CUHII ranged between 0.18 and 2.06°C during HW periods, which is much larger than that during NHW periods varied between 0.03 and 1.32°C. In particular, the average daily CUHII during HW periods exhibited a significant increase of 59.33% compared to that during NHW periods. The maximum ΔCUHII was 0.76°C, occurring at 00:00 BJT, while the minimum ΔCUHII was 0.05°C, observed at 19:00 BJT. It should be noted that the ΔCUHII remained positive throughout the daytime and nighttime, indicating the persistent synergies between HW and CUHI in the built-up area of Beijing.

Fig. 4 illustrates that the ΔCUHII were strongest in 2017, with the ΔCUHII exceeding 0.8°C at six stations in the urban center. Conversely, the weakest ΔCUHII occurred in 2018, with only one station in the urban center recording the ΔCUHII above 0.8°C. Significant spatial variations were observed in the distribution of the ΔCUHII. Regarding the overall pattern of the ΔCUHII across the city, taking the average ΔCUHII as an illustrative example (Fig. 4f), it is evident that the ΔCUHII in the urban north exceeds 0.6°C at five stations, while in the urban south, only four stations record the ΔCUHII above this threshold, indicating the stronger ΔCUHII in the north compared to the south. It is well-documented that the mountain-valley breeze exhibits pronounced wind direction reversal, accompanied by notable differences in wind speeds between the mountain breeze and valley breeze (Zhang et al., 2018; Xue et al., 2023). Furthermore, examining the average ΔCUHII within the built-up area, it is noteworthy that only one station within the Second Ring Road records the ΔCUHII exceeding 0.6°C, with all other stations exceeding this value located beyond the Second Ring Road. Notably, the built-up area within the Second Ring Road in Beijing is predominantly characterized by low-rise, with taller structures concentrated beyond this perimeter (Guo et al., 2024). Consequently, the following analysis will delve into the potential causes of the ΔCUHII pattern in the built-up area of Beijing from two perspectives: mountain-valley breeze and urban morphology.

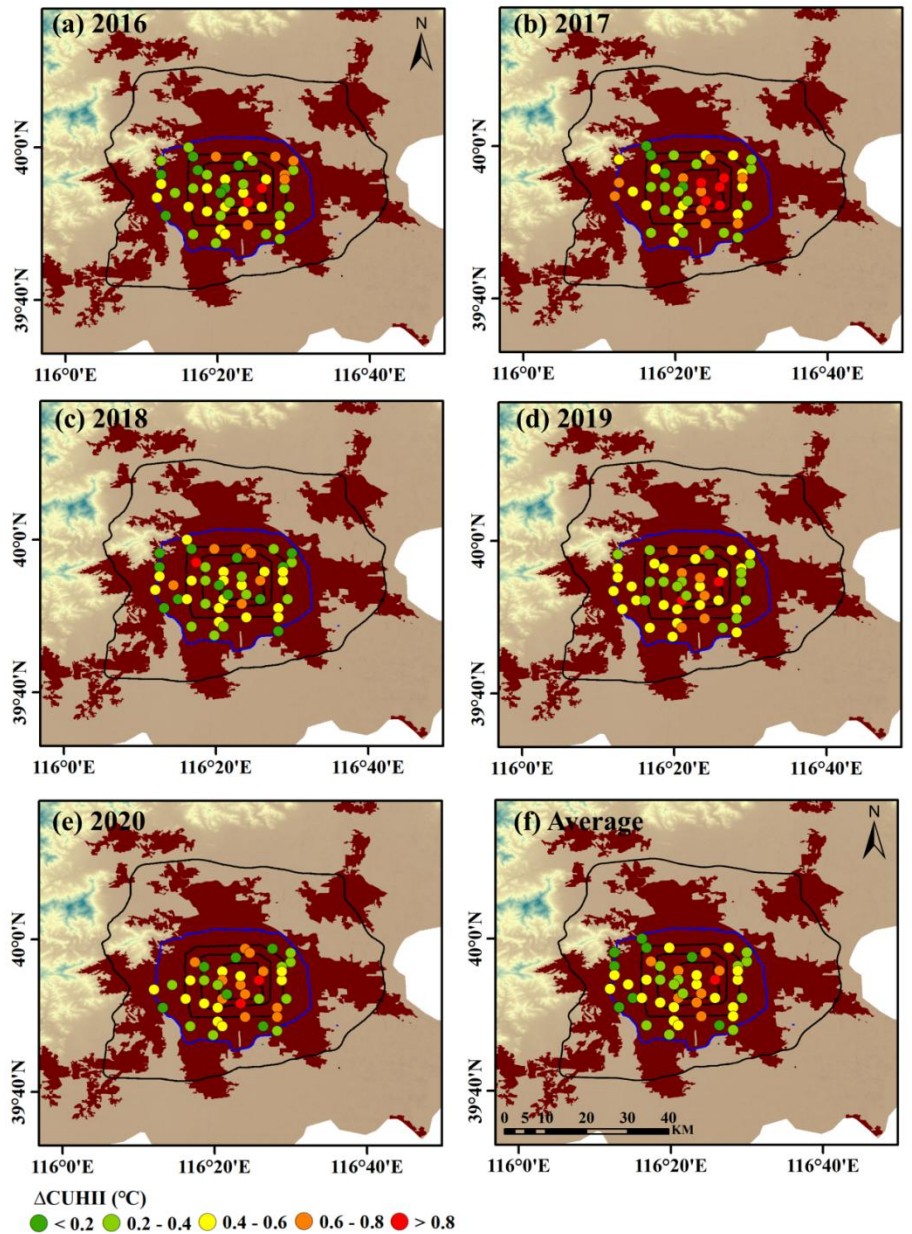

**Figure: 4 Spatial patterns of daily average ΔCUHII based on the urban stations in Beijing during HW periods. (a) 2016, (b) 2017, (c) 2018, (d) 2019, (e) 2020, (f) average value from 2016 to 2020. Different colored dots represent different ranks of the ΔCUHII.**

## 3.2 Modulation of the ΔCUHII by mountain-valley breeze

Local circulations caused by different geographical environments have a significant impact on the spatial and temporal distribution of urban extreme high temperatures (Zhang et al., 2011; Zhou et al., 2020; Chen et al., 2022). The western and northern parts of Beijing are surrounded by mountains, and the mountain-valley breeze strongly impacts the near-surface thermal dynamic field of the Beijing megacity (Miao et al., 2013; Dou et al., 2014). In this section, this research analyzed the modulation of the synergies between HW and CUHII by the mountain-valley breeze using wind field and temperature data from AWS.

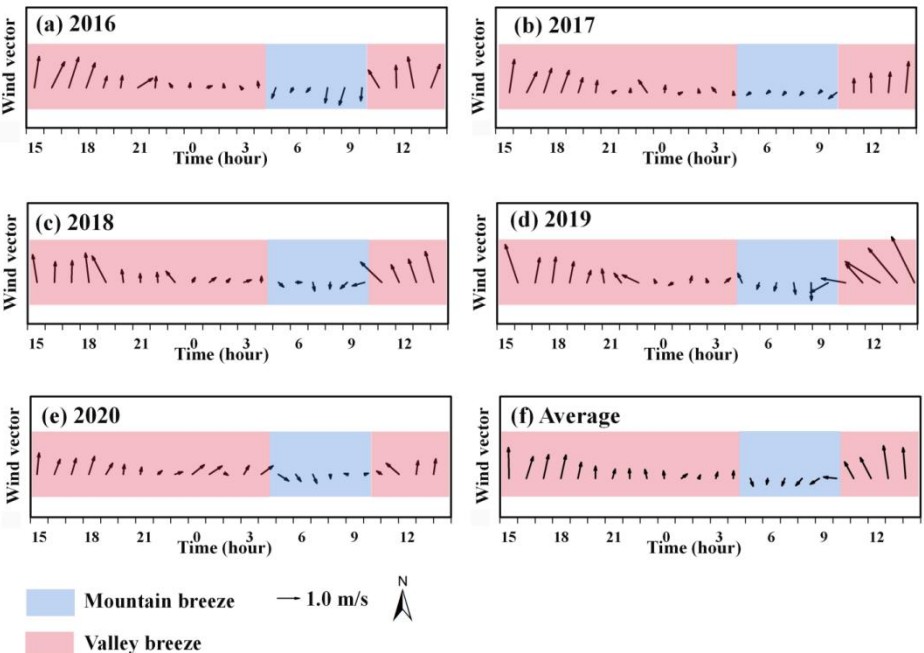

**Figure: 5 Diurnal variations (Beijing time, BJT) in wind direction and wind speed of the reference stations in Beijing during HW periods. (a)2016, (b)2017, (c)2018, (d)2019, (e)2020, (f) average value from 2016 to 2020.**

Based on previous research, it is well-established that there exists a pronounced wind direction reversal between the mountain breeze phase and the valley breeze phase, characterized by significant differences in wind speeds (Jiang et al., 1994; Fu, 1997; Dong et al., 2017). To mitigate the influence of the urban environment disturbing the surface wind measurement, this paper first analyzed the diurnal variation of mountain-valley breeze solely using observation data from reference stations. As depicted in Fig. 5, the reference stations were dominated by northerly winds from 05:00 BJT to 10:00 BJT, with a notable reversal in wind direction occurring at 11:00 BJT, resulting in south winds dominating the reference stations until 04:00 BJT of the following day. The mountain breeze persisted for approximately 5 hours, while the valley breeze lasted for approximately 19 hours. The average speed of the mountain breeze (0.40 m/s) was significantly lower than that of the valley breeze (0.72 m/s), consistent with a previous study (Zheng et al., 2018b). This phenomenon indicated the

significant presence of mountain-valley breeze in Beijing during summer. Based on the statistics presented above, we further analyzed observational data from urban stations and found that the average ΔCUHII during the mountain breeze phase (0.53°C) was higher than that during the valley breeze phase (0.36°C). The effectiveness of urban natural ventilation is contingent upon the exchange and flow of air within the urban canopy layer, which, in turn, exerts a direct influence on the high-temperature environment prevalent within cities (Yang et al., 2023). Consequently, we explored whether the observed

discrepancy in the ΔCUHII between the mountain breeze phase and valley breeze phase was potentially linked to the wind speed difference between these two breeze phases. Fig. 6a-6b illustrates that, regardless of whether it was the mountain breeze phase or the valley breeze phase, the correlations between the wind speed and the ΔCUHII were both negative. Except for the mountain breeze phase in the urban south, the other p-values were lower than 0.1. In the future, we plan to expand our research area to encompass the Beijing-Tianjin-Hebei urban agglomeration. Low wind speeds typically result in

poorer urban ventilation environments (Ng, 2009; Bady et al., 2011), especially in areas with densely packed urban buildings that hinder the flow of cold air. With reduced airflow and limited heat dispersion under weak wind conditions, these conditions further exacerbate urban excess warming (Gemechu et al., 2022).

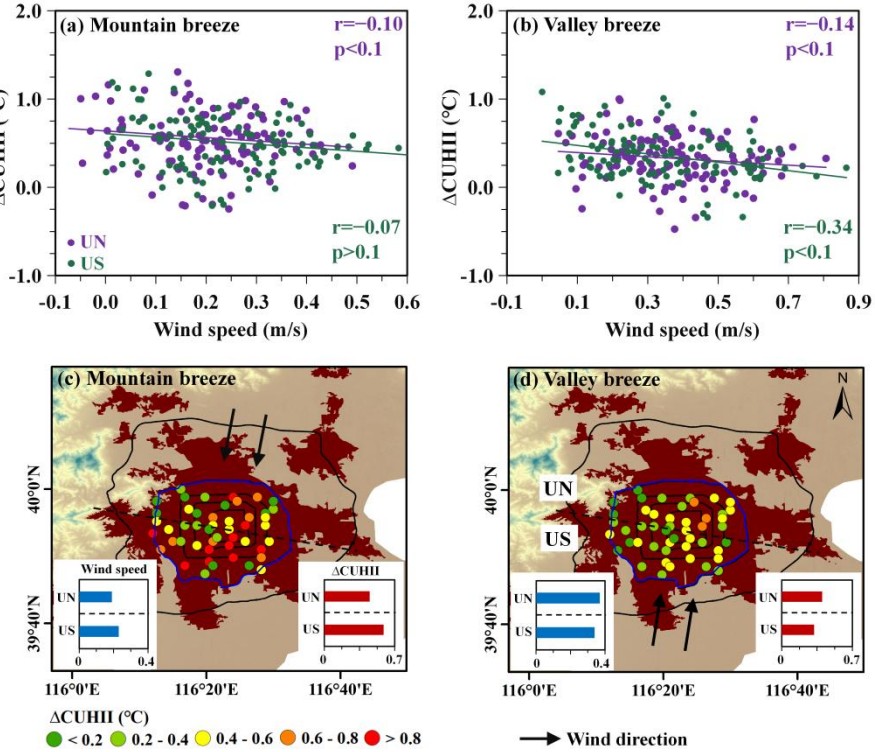

**Figure: 6** Correlation analysis between the wind speed and the ΔCUHII at urban stations from 2016 to 2020 during the mountain
breeze phase (a) and the valley breeze phase (b). The spatial patterns of the ΔCUHII during the mountain breeze phase (c) and valley breeze phase (d). The histogram in the lower left corner represents the average wind speed in the urban north (UN) and urban south (US), while the histogram in the lower right corner represents the average ΔCUHII in the UN and US.

We analyzed the spatial patterns of the ΔCUHII in the urban north (UN) and urban south (US) (as shown by the black dashed line in Fig. 6c-6d). During the mountain breeze phase, the wind direction was from north to south. As shown in Fig. S1, in the urban north, the year with the highest average ΔCUHII was 2016, and the urban south experienced its maximum ΔCUHII in 2018. As shown in Fig. 6c, despite the slightly higher average wind speed in the urban south (0.23 m/s) compared to that in the urban north (0.19 m/s), the annual average ΔCUHII in the urban south (0.57°C) was higher than that in the urban north (0.48°C). During the valley breeze phase, the wind blew from south to north. In Fig. S2, the year with the highest average ΔCUHII in the urban north was 2017. In the urban south, the maximum average ΔCUHII occurred in 2020. In Fig. 6d, although the average wind speed in the urban north (0.38 m/s) was higher than that in the urban south (0.34 m/s), the annual average ΔCUHII in the urban north (0.40°C) was higher than that in the urban south (0.32°C). On an urban scale, it was evident that wind speed might not be the primary regulatory factor for urban excess warming.

According to the above analysis, on the street scale, wind speed was inversely proportional to the ΔCUHII at individual stations, suggesting that favorable ventilation conditions could enhance the thermal environment surrounding specific locations. On the urban scale, the wind direction of the mountain-valley breeze might induced a north-south asymmetrical pattern of urban excess warming during HW periods. Below, we will proceed to analyze the driving effects of urban morphology on the synergies between HW and CUHI.

### 3.3 Response of the ΔCUHII to urban morphology

The spatial heterogeneity of urban areas and their infrastructure can directly contribute to the spatially inhomogeneous distribution of the near-surface air temperature (Fenner et al., 2017). In this section, we further explored the driving factor of the synergies of HW and CUHII in Beijing, focusing on urban morphology.

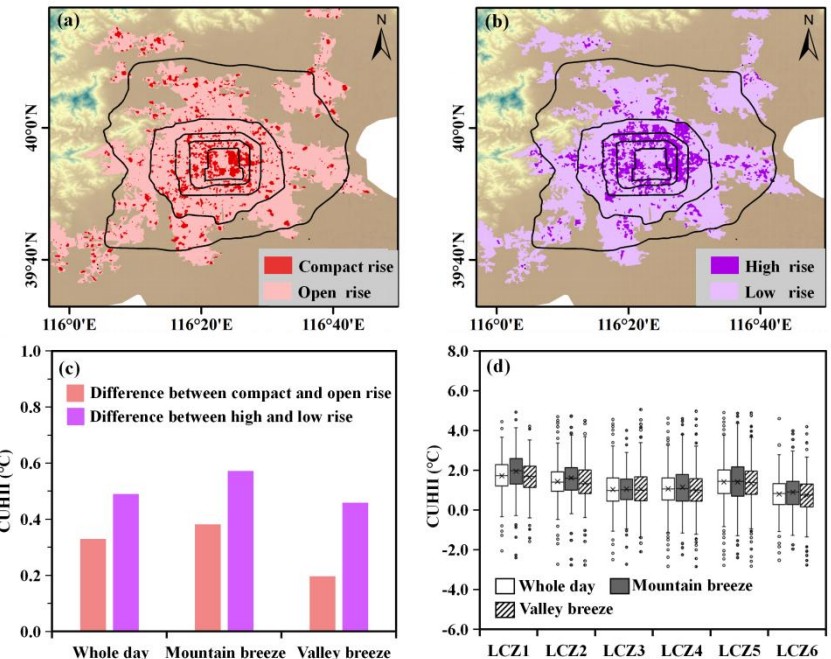

**Figure: 7 (a) Urban configuration structures are dominated by density information, including compact rise (LCZ1, LCZ2, LCZ3) and open rise (LCZ4, LCZ5, LCZ6). (b) Urban configuration structures are dominated by height information, including high rise (LCZ1, LCZ2, LCZ4, LCZ5) and low rise (LCZ3, LCZ6). (c) Differences in CUHII between compact rise and open rise, and between high rise and low rise. (d) Box-and-whisker plots comparing the CUHII values for various LCZs.**

From the perspective of urban configuration structures (Fig. 7a-b), compact rise were mainly concentrated within the Second Ring Road of the built-up area, while high rise was primarily distributed between the Second and Fourth Ring Roads. Notably, most of the stations with high urban excess warming were located in areas with high rise. Fig. 7c demonstrates that the difference in CUHII between compact rise and open rise ranged from 0.20~0.39℃, while the difference between high rise and low rise was 0.46~0.57℃. Previous studies have shown that in densely populated high rise areas of Beijing, HW

events occur more frequently and last longer (Zong et al., 2021). Similar results are obtained in this study. Among various LCZs, LCZ1, which was dominated by compact high rise, had the highest average CUHII value in the built-up area of Beijing, while LCZ6, which was dominated by open low rise, had the lowest average CUHII value in the built-up area of Beijing (Fig. 7d). Therefore, apart from local circulation patterns, the CUHII was also dependent on the characteristics of the urban morphology.

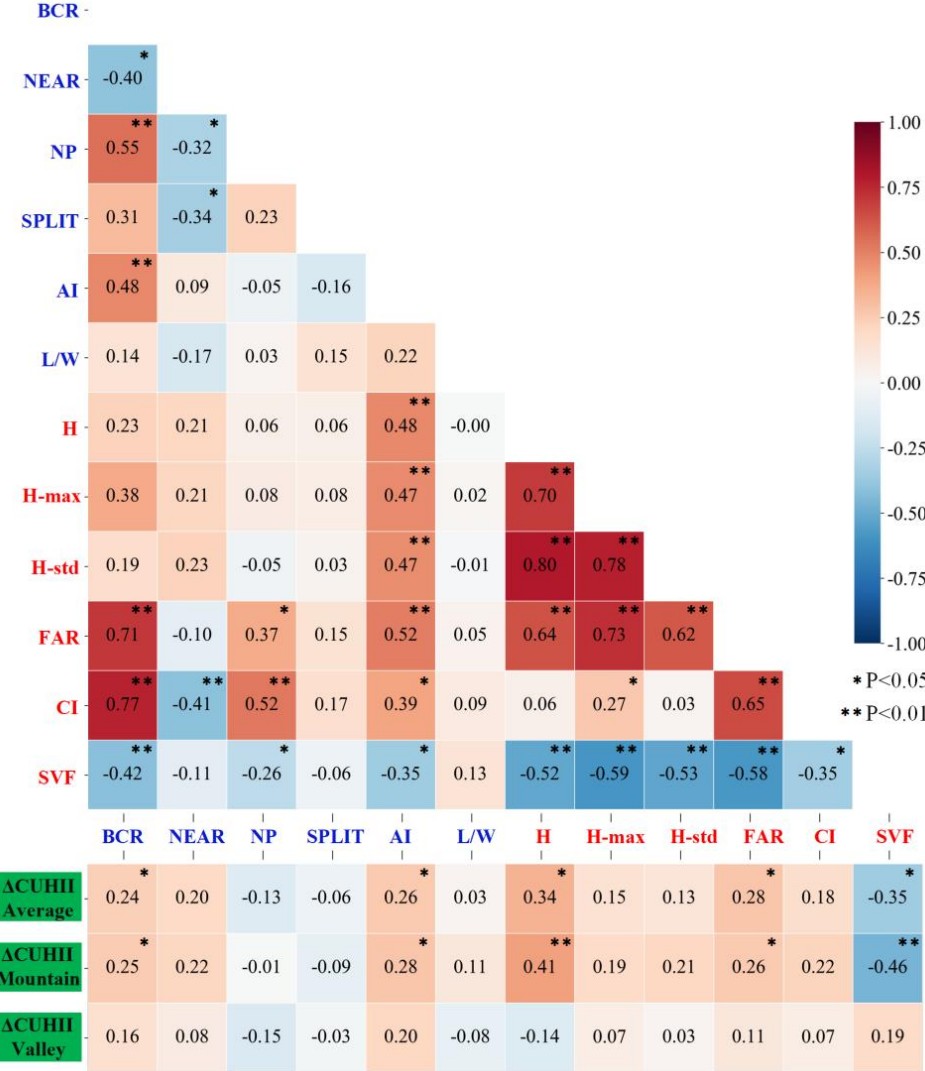

**Figure: 8 Spearman rank correlation coefficients between the urban morphology indicators and the ΔCUHII during different local circulation phases. The blue characters represent the 2D urban morphological indicators, while the red characters represent the 3D urban morphological indicators. The color legend on the right represents the value of correlation coefficients, with blue indicating a low correlation and red indicating a high correlation.**

The Spearman correlation analysis showed that the associations between the 3D indicators and the ΔCUHII were generally higher than those between the 2D indicators and the ΔCUHII (Fig. 8). Indicators using a combination of morphological aspects generally had stronger correlations with temperature (Tian et al., 2019). For example, SVF had the highest correlations with the ΔCUHII among 3D indicators. The correlation between the floor area ratio (FAR) and the ΔCUHII was stronger than that between the building cover ratio (BCR) and the ΔCUHII. Furthermore, the correlations between the 2D indicators and the ΔCUHII, as well as those between the 3D morphological indicators and the ΔCUHII, varied significantly

during different local circulation phases. During the mountain breeze phase, the relationship between the urban morphological indicators and the ΔCUHII was stronger, whereas during the valley breeze phase, this relationship was weaker.

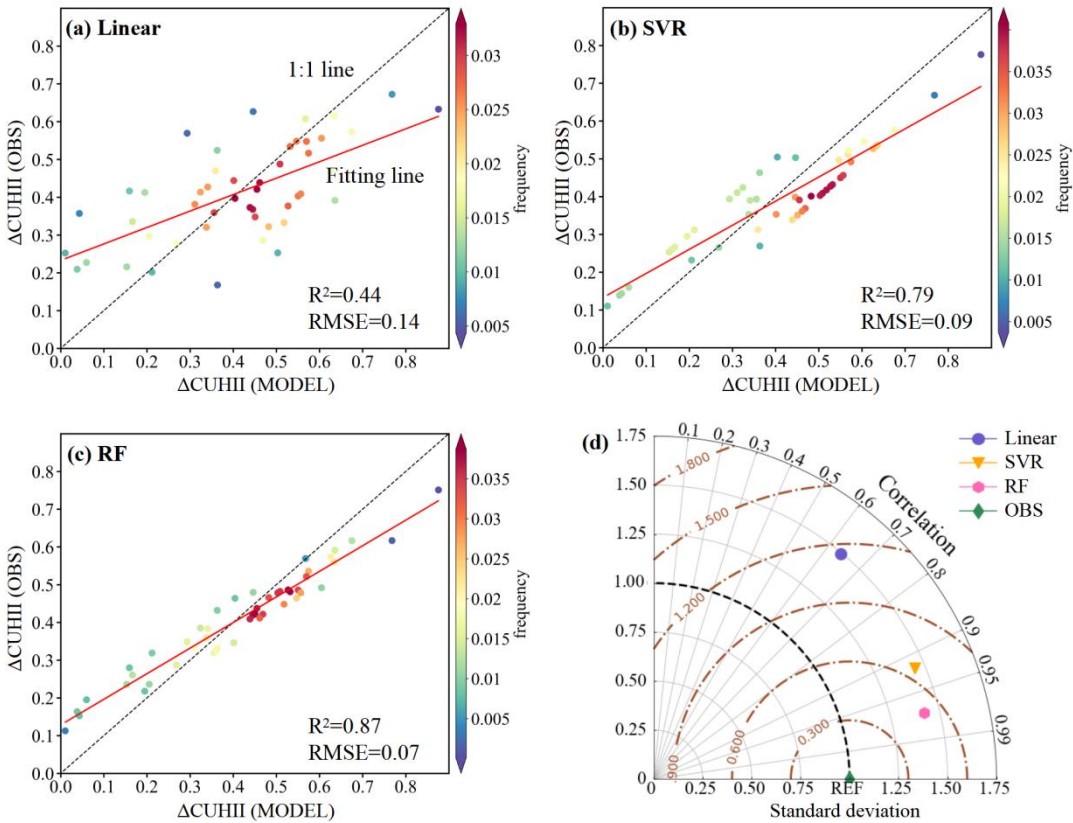

**Figure: 9 Comparing the simulation accuracy across different models. (a) Linear model, (b) SVR model, (c) RF model. ΔCUHII (OBS) represents the observed ΔCUHII values, while ΔCUHII (MODEL) represents the modeled ΔCUHII values. The color legend on the right represents the frequency of sample occurrence. (d) Taylor diagrams for various models, where the gray line represents the correlation between the simulated and observed values, and the brown dashed line indicates the root mean square error between the simulated and observed data.**

As depicted in Fig. 9a, the linear model yielded a coefficient of determination ($R^2$) of 0.44 and a root mean square error (RMSE) of 0.14°C, indicating a relatively large modeling error. Consequently, while the linear model provided a foundational framework for modeling the ΔCUHII, it might not be the most optimal choice for our study. Fig. 9b illustrated

that the SVR model demonstrated superior performance compared to the linear model, achieving an $R^2$ value of 0.79 and an RSME value of 0.09°C. Moreover, the RF model was used to explain the contribution of each feature to the modeling of the ΔCUHII. Based on the performance values given in Fig. 9c, it appeared that RF had the highest $R^2$ value of 0.87 and the lowest RMSE value of 0.07°C, which indicated that it had the lowest modeling error and was potentially more accurate than other models. In Fig. 9d, the gray ray in the Tylor diagram indicated that the correlation between the data from the linear

model and the observed data was relatively low. Additionally, the variation in the linear model data was significantly greater than the observed variation (indicated by the excessive distance from the origin). In both cases, this results in a relatively large, centered root mean square error (yellow contour line) for the linear model. These results suggested that the performance of the linear model was relatively poor. The RF data were 87% correlated to the real data, while the linear and SVR data had a weaker correlation with the real data. The good performance of RF could be proved by its strong correlation with the real data set. Therefore, the RF model could be considered a reliable tool for fitting the relationship between the ΔCUHII and the urban morphology.

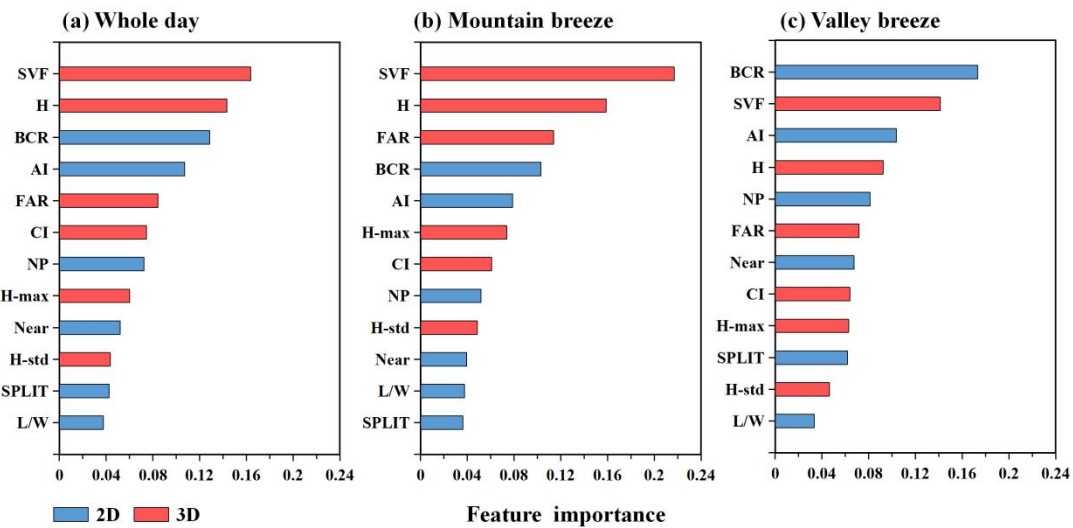

**Figure: 10 The feature importance rank of urban morphological indicators for the RF model estimating the ΔCUHII. (a) Whole day, (b) mountain breeze phase, (c) valley breeze phase. The blue bars represent 2D urban morphological indicators, while the red bars represent 3D urban morphological indicators.**

This paper constructed an RF model to compare the relative importance of urban morphology in modeling the ΔCUHII. The importance of indicators varied by different local circulation phases. Throughout the whole day (Fig. 10a), the relative importance was listed in descending order of importance: SVF, FAR, H, BCR, CI, AI, NP, H-max, Near, H-std, SPLIT, L/W. During the mountain breeze phase (Fig. 10b), despite the alteration in the order of importance of indicators, the SVF was still the most important morphology indicator for modeling the ΔCUHII. Previous studies have shown that SVF is closely related to urban land surface temperature (LST) (Peng et al., 2017; Scarano & Mancini, 2017) and air temperature (Rafiee et al., 2016; Drach et al., 2018). Compared to the immediately neighboring rural area, SVF played a more important influence on determining the LST in the high rise of the built-up area (Jia et al., 2023). During the valley breeze phase (Fig. 10c), the importance of SVF to the ΔCUHII has weakened, ranking second in the importance list. H and SVF had weaker correlations with daytime temperature but showed stronger correlations with nighttime temperature (Tian et al., 2019). Overall, the

importance list showed that the effects of 3D indicators on the ΔCUHII were stronger than the effects of 2D indicators on the ΔCUHII.

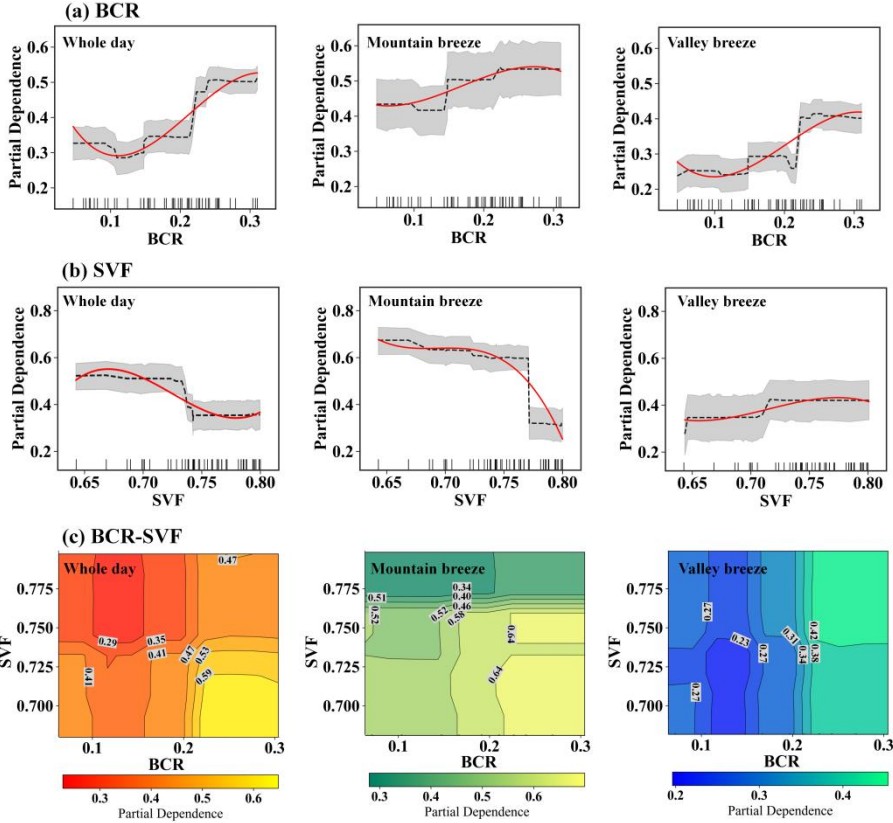

Figure: 11 (a-b) Partial dependence plots of the ΔCUHII on BCR and SVF. The red line represents the fitted curve, while the gray lines indicate the 95% confidence interval. The rug plots (small vertical lines) along the X-axis represent the distribution of the feature values. (c)The two-way plots partial dependence of the ΔCUHII on BCR and SVF. The X-axis represents the BCR feature, while the Y-axis represents the SVF feature. The interpolated colors of the panel, ranging from dark to light, signified the partial dependence decreasing from large to small.

As previously demonstrated, the importance of SVF and BCR in the 3D and 2D indicators was the highest. Partial dependence, in the context of machine learning, refers to the assessment of the relationship between a single feature and the model's predicted outcome, while all other features in the dataset are held constant (Friedman, 2001). This function represents the effect of selected explanatory variables and can be used to interpret "black box" models (Cutler et al., 2007; Shiroyama & Yoshimura, 2016). In Fig. 11a, it could be seen that as the BCR increased in summer, the ΔCUHII showed a continuous upward trend overall. The growth trend of the ΔCUHII during the mountain breeze phase was higher than that during both the whole day phase and the mountain-valley breeze phase. When the BCR exceeded 0.1, the dependence of the ΔCUHII on the BCR increased rapidly. There might be a threshold for the building area, and when this threshold was exceeded, the promoting effect of the building area on the ΔCUHII was significantly enhanced. This complex pattern of

395 association is closely related to urban climatic conditions, vegetation coverage in the built-up area, the frequency of human activities, and seasonal and spatial differences in energy consumption (Guo et al., 2016; Yang et al., 2018; Zhou et al., 2014). In Fig. 11b, during both the whole day phase and the mountain breeze phase, as SVF increased in summer, the overall synergistic effect exhibited a continuous downward trend. However, during the valley breeze phase, the dependence increased with increasing SVF, indicating that SVF had an inhibitory effect on the ΔCUHII. In addition, the two-way partial

dependence plots were constructed to explore the joint effect of two dominant factors (Fig. 11c). The interactions between BCR and SVF relied on their relative values. During both the whole day phase and the mountain breeze phase, the peak partial dependence of the ΔCUHII was observed in regions characterized by BCR values exceeding 0.2 and SVF values less than 0.72. During the valley wind phase, the region demonstrating the highest ΔCUHII dependence continued to be located within areas where BCR exceeds 0.2, but the SVF value was significantly higher than 0.72, showing that SVF had a dual

impact on the ΔCUHII.

The preceding analysis demonstrated that the RF model served as a reliable tool for simulating the relationship between the ΔCUHII and the urban morphology. The importance ranking highlighted that 3D morphological indicators exerted a more significant influence on the ΔCUHII compared to 2D morphological indicators. Notably, the impact of SVF on the ΔCUHII was intricate. In the following discussion section, we will select representative sites for further investigation.

**4 Discussions**

Local circulations and urban morphology played pivotal roles in influencing the ΔCUHII. In the following, this article selected representative stations with typical geographic locations and spatial characteristics of buildings to analyze how local circulations and urban morphology alter the ΔCUHII.

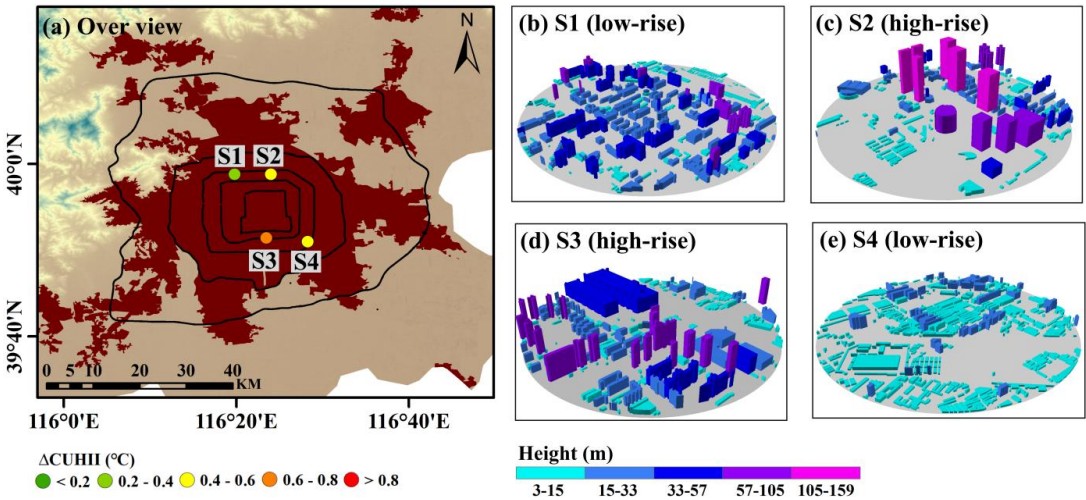

Figure: 12 (a) An overview of representative urban stations in the built-up area of Beijing. (b-e) Urban morphology around the representative stations. The different colors on the buildings represent their respective heights.

Taking into account the influence of the mountain-valley breeze, representative stations were selected in the urban south and north in this section. Additionally, based on the driving effects of urban morphology, we select high rise and low rise as the criterion of representative stations. Ultimately, 651061 (S1), 651007 (S2), 651047 (S3), and 651009 (S4) were chosen as representative stations (Fig. 12). S1 and S2 were located between the Third and Fourth Northern Rings, with S1 mainly surrounded by low rise and S2 surrounded by high rise. Meanwhile, S3 and S4 were situated between the Third and Fourth Southern Rings, with S3 mainly surrounded by high rise and S4 surrounded by low rise. The comparison of the ΔCUHII difference between stations in the urban north (S1 and S2) and stations in the urban south (S3 and S4) could be utilized to study the impact of mountain-valley breeze on the ΔCUHII. Furthermore, contrasting the ΔCUHII difference between stations surrounded by low rise with larger SVF (S1 and S4) and stations surrounded by high rise with smaller SVF (S2 and S3) provided an opportunity to analyze the influence of urban morphology on the ΔCUHII.

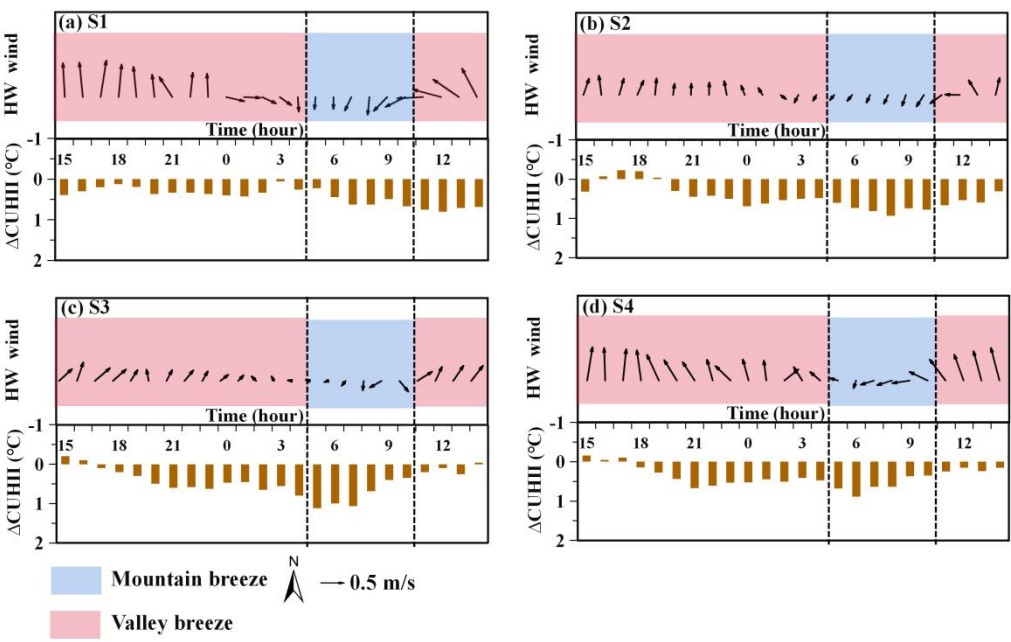

**Figure: 13 Diurnal variations in wind direction, wind speed, and the ΔCUHII in the built-up area of Beijing during HW periods. The blue boxes represent the mountain breeze phase, while the red boxes represent the valley breeze phase.**

During the mountain breeze phase, the wind direction is from north to south. As depicted in Fig. 13, the observed ΔCUHII in the urban north (S1 at 0.51°C, S2 at 0.76°C) was lower than that in the urban south (S3 at 0.77°C, S4 at 0.59°C). For the entire city, a more consistent wind field at the ground level results in a stronger heat transport capacity (Xie et al., 2022; Yang et al., 2023). Combining with the statistical results in Fig. 6c-6d, we suggested that the direction of the mountain-valley breeze might alter the pattern of the ΔCUHII across the entire urban area. In the next, we examined the influence of urban morphology on the ΔCUHII. Using urban north stations (S1 and S2) as examples, S2 (surrounded by high rise) exhibited

stronger ΔCUHII than S1 (surrounded by low rise). High rise residential buildings are associated with higher population densities with greater capacities to mitigate heat, translating to more air conditioners which when operating release additional heat (Ryu & Baik, 2012). High rise neighborhoods have smaller SVF and thus have less outgoing long-wave radiation (Unger, 2004). High rise with smaller SVF neighborhoods tend to experience lower wind speeds (Hang et al., 2011). The lower wind speed limited the loss of sensible heat through atmospheric convection and advection, making it difficult for heat to dissipate from the streets (Wang et al., 2009). During the mountain breeze phase, the SVF of buildings primarily exhibited an enhancing effect on the ΔCUHII.

During the valley breeze phase, the wind direction is from south to north. During the valley breeze phase, the wind direction is from south to north. The ΔCUHII observed at S1 (0.35℃) and S2 (0.34℃) located in the urban north was greater than that at S3 (0.31℃) and S4 (0.32℃) located in the urban south. Similarly, this paper considered that this pattern might be related to the influence of large-scale horizontal heat transport. In terms of urban morphology, between 11:00 BJT and 18:00 BJT, the ΔCUHII at S3 (surrounded by high rise) was 0.01°C lower than that at S4 (surrounded by low rise), indicating that the inhibitory effect of high rise on the ΔCUHII was dominant. Although high rise can enhance the ΔCUHII by reducing outgoing longwave radiation and wind speed, on the other hand, they block more shortwave solar radiation from reaching the ground, and their shading effect contributes to a decrease in near-surface air temperature (Zhang et al., 2016; Krayenhoff & Voogt, 2016; Taleghani et al., 2016; Cai, 2017). After sunset (19:00 BJT), the ΔCUHII observed at S3 was 0.07°C higher than that at S4, signifying that the enhancement of the ΔCUHII by high rise reasserted its dominance. During the valley breeze phase, the high rise with smaller SVF exerted a dual influence on the ΔCUHII.

**5 Conclusions**

This study selected the Beijing megacity as the research subject, utilizing high-density AWS data from 2016 to 2020 as the research sample. Through remote sensing data and machine learning models, the synergies between HW and CUHI were analyzed.

During HW periods, the average daily CUHII underwent a substantial increase of 59.33% compared to NHW periods. The maximum urban excess warming was observed between the Second and Fourth Rings of Beijing. On an urban scale, the large-scale horizontal heat transport caused by the wind direction reversal of mountain-valley breeze might lead to an asymmetric pattern of the ΔCUHII. On a street scale, the wind speed and ΔCUHII exhibited a negative correlation. Additionally, the impact of urban morphology could not be ignored. The average CUHII of compact high rise (LCZ1) was significantly higher than that of open low rise (LCZ6). The importance order of the RF model indicated that the effects of 3D indicators on the ΔCUHII were stronger than the effects of 2D indicators on the ΔCUHII. In the partial dependence plots, the SVF of buildings exhibited a complex influence on the synergies between HW and CUHI. Ultimately, through the analysis of representative stations, we observed that during the mountain breeze phase, high rise with lower SVF primarily enhanced the ΔCUHII. However, during the valley breeze phase, the effect of high rise with lower SVF on the ΔCUHII was dual. In

the future, we will continue to investigate the mechanism of synergies between HW and CUHI using high-resolution observational data and numerical models, to provide crucial theoretical foundations and technological support for the construction of a comprehensive high-temperature monitoring, forecasting, and warning system.

**Data availability.** The hourly AWS observation data are available upon request from the China Meteorological Data Service Center (http://data.cma.cn/en). The land cover data are available at https://zenodo.org/record/5816591 (Yang & Huang, 2021).

**Author contributions.** Tao, S., Yuanjian, Y. conceptualized the study. Tao, S. wrote the original manuscript and plotted all the figures. Yuanjian, Y., Ping, Q., and Simone, L. assisted in the conceptualization and model development. All the authors contributed to the manuscript preparation, discussion, and writing.

**Financial support.** This study was supported by the National Natural Science Foundation of China (42105147), the Joint Research Project for Meteorological Capacity Improvement (22NLTSQ013), and the Collaborative Innovation Fund of the Education Department of Anhui Province (GXXT-2023-050).

**Competing interests.** The contact author has declared that none of the authors has any competing interests.

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
