# Peer review of "Diurnal variation of amplified canopy urban heat island during heat wave periods in the Beijing megacity: Roles of mountain-valley breeze and urban morphology"

_EGUsphere, 2024_

## Referee Comment (RC1)

**EGUspehre**

**Diurnal variation of amplified canopy urban heat island in Beijing megacity during heat wave periods: Roles of mountain-valley circulation and urban morphology**

**Review**

This study focuses on the canopy urban heat island (CUHI) for the city of Beijing, based on surface weather station observations. It seeks to understand the effects of intensification of the UHI phenomenon during heat waves, and the role of local breeze circulations (mountain and valley) and of urban parameters on the temporal and spatial variability of the intensification.
It is an interesting scientific subject that looks at the urban climate of cities in complex environments, and that fits in well with the target scientific journal. It is addressed through an experimental approach, made possible by a fairly dense surface observation network in the city and surrounding area.

Nevertheless, the central scientific questions of the study are not, in my opinion, presented and structured clearly enough. The article would be clearer and more interesting if these questions were clearly stated and accompanied by a well-structured step-by-step analysis. As it is, the article investigates many different issues (CUHII, the effect of heatwaves, the effect of breeze circulation, the effects of urban parameters, cross-effects, the comparison of statistical approcahes, etc.), and it is sometimes difficult to see the coherence of the whole. And in the end, the main findings don't stand out clearly enough. Also, the data used and the methodologies chosen (as well as the Figures) could be explained more precisely.

In my opinion, this work needs to be reworked in depth and some methodological issues need to be revisited. I do not recommend publication of this article.

**Major comments**

First, the methodologies are not clearly explained, so the reader often lacks elements of understanding:
- For example, there is no clear understanding of the available network, i.e. the location of stations according to urban typologies, the different land use characteristics in and around the city. This should be presented in the section on methods and data.
- The geographical context is presented, but there is a lack of a more complete description of mountain and valley breeze situations (based in particular on the existing literature, which is apparently fairly extensive): the mechanisms involved, the factors of variability in terms of intensity and daily cycle, the influence of HW conditions, etc.
- The method used to calculate CUHI is unclear to me. Is it based on all days of the year vs heatwave days, or on summer days vs heatwave days?

Also, some methodological choices are highly questionable.
1/ the time period 2016-2020 is far too short to be considered as a climatology; a time series of around thirty years is needed to extract HW detection thresholds
2/ some of the stations are located in urban environments, so the maximum daily temperature can potentially be influenced

3/ Again, if some of the stations are located in urban environments, they should not be considered when calculating the synoptic wind (the urban environment disturbs the surface wind measurement). The synoptic wind should be calculated on the basis of rural stations only.
> If the objective here is to identify HWs on a regional/local scale, I suggest applying detection only to non-urban stations (to avoid any urban influence), and selecting stations for which very long time series are available.

Generally all the captions need to be improved, especially the legends which are not detailed enough, so that it can be difficult to understand what is presented.

It seems to me that it is difficult to conclude from these figures about the influence of mountain or valley breezes on the CUHII knowing that the phenomenon of CUHI has a marked diurnal cycle. On the other hand, the effect of the wind on the UHI may be delayed over time: if there was wind during the day, there is less heat accumulation and then possibly less UHI at night.
Same for urban parameters: it's very interesting to see how the different urban parameters rank in terms of their influence on CUHII. However, I wonder about the relevance of comparing this for the "mountain breeze" and "valley breeze" cases. The CUHI phenomenon is different during the day and at night, and is not related to the same physical processes, so it seems difficult to draw relevant conclusions from these comparisons.

**Minor comments**

P3, L69
*"... more than 1,400 km$^2$ ..."*

P3, L71
*"The altitudes of those mountains exceed 2,000 meters."*

P3, L71
*"The northeastern region comprises ..."*

P3, L73
Please clarify what you mean by "weak weather system"

P3, L73-75
It would be interesting to summarise here the main findings of these various studies on breezes and local atmospheric circulations

P4, L82-83
*Land cover modulates the energy exchange, water, and carbon cycle between different regions of the Earth.*
I should remove the second part of the sentence.

P5, L85
*"The annual China Land Cover Dataset (CLCD) is a dynamic data set accounting for land use in China released by  Yang & Huang (2021). They made the land cover datasets with a spatial resolution of 30 m based on 335,709 Landsat images on Google Earth Engine."*

P5, L88
Define the acronym LCZ (local climate zone) which is use here in the text for the first time, and include the ref to Stewart and (2012).

P5, L90
*"... within the research buffer areas of the target stations"*
This information should be introduced later in the text once the stations have been presented.

P5, L94-95
Replace "encompasses" (not really appropriate) and clarify what you mean by "related elements"

P5, L110
*"... otherwise, it was considered* ==as a non heat wave (NHW)==  *day."*
Are NHW days defined for the whole year or just for the summer period?

P5, L111-112
*"... by selecting reference stations for ground temperature observations and urban stations"*
Please clarify this sentence, do you mean "... by selecting urban reference stations for retrieving near-surface air temperature observations"?

P5, L113
*"... located outside of a 50km radius"*
Please add a space between "50" and "km" (and do the same everywhere in the rest of the text)

P5, L112-114
Based on which temperature distribution (day, night, average)? Why not base it on the land use map?

P5, L116
*"... than the average altitude of* ==the== *45 urban stations"*
How is the station's rural environment defined? From the land use map, I presume ?
As I understand it, this means that the other 45 stations (which are not classed as "rural") are all urban, and that they all meet the min distance to the city centre and land use conditions ?

P6, L119-121
You should explain more clearly what is a valley/mountain breeze.

P6, L125
*"... the daily average* ==components of the== *wind U and V were obtained ..."*

Section 2.3.3 and Tab1
Not all morphological indicators are well described or easy to understand. The methodology lacks precision. For example: (1) what does "patch" mean? (2) on which zone are the indicators calculated, is this the case for all? (3) what is the size of the buffer? What are the final results ?

P7, L151-152
What means "The impact of urban spatial morphology on urbanization bias was evaluated" ?

P8, 164-175

The presentation and analysis of Fig 2 are extremely confusing. Personally, I don't understand what is presented here. Is it a difference between urban and rural areas, given that the aim is to study "urban warming excess" ?

P9, L180-189 and Fig. 3
- CHUII values are relatively low for a city like Beijing. We would expect higher intensities, particularly during heatwaves. How do you explain this? And could it have something to do with the methodology?
- I don't think that the variability is any greater during the day than at night. It also follows a plateau during the day (except in the transition phases).
- *"The diurnal variation of CUHII may be modulated by anthropogenic heat emissions, aerosols, atmospheric circulation, etc. (Zheng et al., 2018; Zheng et al., 2020; Yang et al., 2020)."* This sentence is off-topic, there is no link with the discussions of the dirunes cycles.
- Fig.3: I suggest plotting the daily CUHII cycles centred on the night-time hours (when intensities are at their highest).
- Explain in the text what means BJT
- The fact that urban heat islands are stronger during heatwaves is well known. It is based on the physical processes involved and has been observed in a large number of situations/cities.

P12, L194 and Fig. 4
- The times at which the ΔCUHII is calculated are not specified (day, night, daily average). It would make much more sense to separate the hours of day and night, or even focus of nighttime (the phenomenon being nocturnal).
- Fig. 4: This figure presents both the spatial variability of ΔCHUII and the interannual variability. This could be interesting if the analyzes were a little more in-depth. Here there are no very clear conclusions/messages about the influence of e.g. urban morphology or the variability of synoptic conditions.

P12, L205
*"In this section, this research analyzed the modulation of mountain-valley breeze on the synergies between HW and CUHI..."* > *"... the modulation of the synergies between HW and CUHI by the mountain-valley breeze"* ?? This is what you mean ?

P15, L242-244
I don't understand on what basis this comment is made

Section 3.3
The city configuration with variability of building densities and heights is interesting.

P 16, Fig. 7c
According to the figure, D-value (for dense vs open) is stronger for "whole days" case than for both "mountain breeze" and "valley breeze" cases. However, the D-value for "whole days" should be intermediate to the other two cases, if it's calculated as an average over all the hours of the day, right?

P17, L287
What do you mean by "during 3D indicators" ? I don't understand this sentence.

P17, L287

The term "amplified CHUII" could be clearly explained once and then replaced by ΔCUHII (everywhere in the text and figures).

P18, L289-291
You say *"Urban morphological indicators had weaker relationships with amplified CHUII during the mountain breeze period but showed stronger correlations with amplified CHUII during the valley breeze period."*
According to Fig. 8, the effect of urban indicators on ΔCUHII according to the breeze circulations is the opposite of what is written here: the effect is stronger during mountain breezes.

Fig. 9
We don't understand what is presented here, the names of the axes are not explicit and the legend is not detailed enough. I presume it is ΔCUHII(OBS) vs ΔCUHII(MODEL) ?

P18, 297-303
- Linear model: It's rather debatable to say here that the linear model is good (especially as you go on to say that it doesn't perform well...). The RMSE of 0.14°C is rather high given the average values. > you should adapt your comments
- SVR and RF: both models overestimate high values and underestimate low values, why ?

P20, L331
*"... the importance of SVF and BCR in the  indicators ..."*

Fig. 11
- Again we don't understand what is presented here, what is the partial dependence ? clarify what is presented both in the text and in the caption. >> Is it ΔCUHII on y-axis ?
- The range of y-axis is different in the different plots, you should use the same.
- What do the small vertical lines on the x-axis represent?
- For (c) panel, do you think it is relevent to interpolate the data ? I would use symbols instead. Also you should add a color legend

P22, L370-372
On what basis do you say here that the synergistic effect observed at S1 and S2 is lower than that observed at S3 and S4 ? The differences are considered as significative for ex. betwwen S2 and S4 ?

---

## Author Comment (AC1)

**Response to Reviewer Comments**

Dear Reviewer and Editors:

We are sincerely grateful to the editor and reviewer for their valuable time for reviewing our manuscript. The comments are very helpful and valuable, and we have addressed the issues raised by the reviewer in the revised manuscript. Please find our point-by-point response (in blue text) to the comments (in black text) raised by the reviewer. We have revised the paper according to your comments (highlighted in red text of the revised manuscript).

Sincerely yours,

Dr. Yuanjian Yang, representing all co-authors

**Reviewer #1:**

**This study focuses on the canopy urban heat island (CUHI) for the city of Beijing, based on surface weather station observations. It seeks to understand the effects of intensification of the UHI phenomenon during heat waves, and the role of local breeze circulations (mountain and valley) and of urban parameters on the temporal and spatial variability of the intensification. It is an interesting scientific subject that looks at the urban climate of cities in complex environments, and that fits in well with the target scientific journal. It is addressed through an experimental approach, made possible by a fairly dense surface observation network in the city and surrounding area.**

**Nevertheless, the central scientific questions of the study are not, in my opinion, presented and structured clearly enough. The article would be clearer and more interesting if these questions were clearly stated and accompanied by a well-structured step-by-step analysis. As it is, the article investigates many different issues (CUHII, the effect of heatwaves, the effect of breeze circulation,**

the effects of urban parameters, cross-effects, the comparison of statistical approcahes, etc.),and it is sometimes difficult to see the coherence of the whole. And in the end, the main findings don't stand out clearly enough. Also, the data used and the methodologies chosen (as well as the Figures) could be explained more precisely.

*Response:* Thanks very much for taking time out of your busy days to provide us with such valuable comments that significantly improve the quality of our manuscript. In line with your comments and suggestions, we have revised our manuscript carefully and prepared a list of point-by-point responses below.

Firstly, the Introduction section is updated carefully to better highlight the central scientific questions of this study. In our revised manuscript, we focus on the scientific question of how mountain-valley breeze and urban morphology drive the amplified CUHII during HW periods (ΔCUHII) in the Beijing megacity.

Secondly, we have revised the descriptions of the key concepts and calculation methods for CUHII, HW, mountain-valley breeze, and urban morphology in the Data and Methodology section. These updates ensure that readers have a clear understanding of the approach in this manuscript.

Additionally, a summary of the key findings and the corresponding subsequent analysis are appended at the end of each subsection of the section of Results to improve the coherence of this study.

Finally, we have thoroughly revised the Abstract and Conclusion sections to highlight our main findings. Specifically, we have emphasized the influence of wind speed and direction on the temporal and spatial distribution of the positive feedback effect between HW and CUHII, as well as the driving role of both 2D and 3D indicators of urban morphology in the effect.

You can find the details in our revised manuscript.

**Major comments:**

**1.There is no clear understanding of the available network,i.e. the location of stations according to urban typologies, the different land use characteristics in and around the city. This should be presented in the section on methods and data.**

*Response:* Thanks very much for your valuable comment. In our revised manuscript, the details of the method for identifying urban stations and reference stations are added in the section of Data and Methodology.

In general, the CUHII is defined as the temperature difference between the urban station and the rural reference station (Ren et al., 2007; Shi et al., 2015). Thus, identifying the urban stations and rural reference stations is very important for investigating urban climate. In the region of our study, Beijing has undergone massive and rapid urbanization, with its urban space continually expanding into the suburbs over the past few decades. Currently, Beijing boasts a population of 20 million and a built-up area spanning 1400 km². Due to this expansion, a swift transportation system has become imperative for urban development, prompting Beijing to commence the construction of a Multiple-ring-road system since the 1990s (Wang et al., 2010). These rings effectively represent the radial expansion of urban zones, with varying population and building densities. Notably, the Fifth Ring Road, with a length of 98.6 km and a built-up area of approximately 300 km² (depicted as the blue ring in Fig. R1), encompasses the primary regions of the built-up area (Yang et al., 2013). The distribution characteristics of average air temperature around the built-up area of Beijing are illustrated in Fig. R1, showing that the high-temperature zone in the city center aligns closely with the extent of the Fifth Ring Road. Additionally, the proportion of densely built-up areas within the Fifth Ring Road exceeds 85%, significantly higher than that outside this ring. Therefore, we have designated stations within the Fifth Ring Road as urban stations in this study.

The identification of reference stations is shown below. Firstly, the reference stations should have significantly lower temperatures than those of urban stations, based on the spatial distribution characteristics of average temperatures. Secondly, the

reference stations must be located more than 50 km away from the city center, in a rural environment, predominantly situated within areas of sparse trees and shrubs (Yang et al., 2023). Thirdly, the reference stations should also be evenly distributed across different directions of the entire city. According to these criteria, eight reference stations were selected (green plot in Fig. R2), with an average altitude of 39.6 m, which is only 8.8 m lower than the average altitude of 45 urban stations (red plot in Fig. R2).

[Figure]

**Fig. R1 The distribution characteristics of average air temperature around the built-up area of Beijing.**

[Figure]

**Fig. R2 Spatial distribution of urban stations and reference stations in Beijing.**

**Reference:**

Yang, P., Ren, G., Liu, W.: Spatial and temporal characteristics of Beijing urban heat island intensity. Journal of Applied Meteorology and Climatology, 52, 8, 1803-1816, http://doi.org/10.1175/JAMC-D-12-0125.1, 2013.

Ren, G., Chu, Z., Chen, Z., Ren, Y.: Implications of temporal change in urban heat island intensity observed at Beijing and Wuhan stations, Geophysical Research Letters, 34, 5, https://doi.org/10.1029/2006GL027927, 2007.

Shi, T., Huang, Y., Shi, C., & Yang, Y.: Influence of Urbanization on the Thermal Environment of Meteorological Stations: Satellite-observational Evidence, Advances in Climate Change Research, 1, 7–15, https://doi.org/10.1016/j.accre.2015.07.001, 2015.

Yang, Y., Guo, M., Wang, L., Zong, L., Liu, D., Zhang, W., Wang, M., Wan, B., Guo, Y.: Unevenly spatiotemporal distribution of urban excess warming in coastal Shanghai megacity, China: Roles of geophysical environment, ventilation and sea breeze, Building and Environment, 235, https://doi.org/10.1016/j.buildenv.2023.110180, 2023.

Wang, X., Li, X., Feng, Z.: Research on Beijing urban expansion based on the principle of information entropy. China Population,Resources and Environment, S1, 88–92, https://doi.org/CNKI:SUN:ZGRZ.0.2010-S1-024, 2010.

**2.The geographical context is presented, but there is a lack of a more complete description of mountain and valley breeze situations (based in particular on the existing literature, which is apparently fairly extensive): the mechanisms involved, the factors of variability in terms of intensity and daily cycle, the influence of HW conditions, etc.**

*Response:* Thanks very much for your valuable comment. In response to your feedback, we have provided a more comprehensive explanation of the mechanisms, varying factors, and potential impacts of mountain-valley breeze in our revised manuscript as below.

Line 41-55 in the introduction:

[revised manuscript text omitted]

**3.The method used to calculate CUHI is unclear to me. Is it based on all days of the year vs heatwave days, or on summer days vs heatwave days?**

***Response:*** Thank you for bringing this clarification to our attention. The method used to calculate CUHII was specifically based on comparing the air temperature differences between urban stations and reference stations during the summertime.

$$CUHII = T_{urban} - T_{reference} \quad (1)$$

CUHII is the canopy urban heat island intensity during the summertime, $T_{urban}$ is the air temperature of the urban stations, and $T_{reference}$ is the summer air temperature of the reference stations. Based on the above method, the diurnal variation of CUHII is also calculated in this study and is shown in Fig. R3. In Fig. R3, the blue line represents the diurnal variation of summer temperature at the urban station, while the green line depicts the diurnal variation of temperature at the reference station. By calculating the difference between these two stations, we obtained the diurnal variation of CUHII during the summertime.

[Figure]

**Fig. R3 The diurnal variation of CUHII in the built-up areas of Beijing during the summertime.**

**4.The time period 2016-2020 is far too short to be considered as a climatology; a time series of around thirty years is needed to extract HW detection thresholds. Some of the stations are located in urban environments, so the maximum daily temperature can potentially be influenced. If the objective here is to identify HWs on a regional/local scale, I suggest applying detection only to non-urban stations (to avoid any urban influence), and selecting stations for which very long time series are available.**

*Response:* Thank you for your thoughtful comments and suggestions. You are absolutely right that short-term observational data may not fully capture the regional climate characteristics. However, due to the constraints of data access permissions and the use of high-density automatic weather stations, we were unable to obtain 30 years of observational data for our study. Thus, we have adopted the HW criteria published by the China Meteorological Administration in line 133-148 of the revised manuscript:

"Due to variations in climatic backgrounds, geographical conditions, socioeconomic factors, and other variables, different standards have been adopted for studying HW events across the world. The World Meteorological Organization suggests that an HW event occurs when the daily maximum temperature exceeds 32°C and persists for more than three consecutive days. The National Oceanic and Atmospheric Administration of the United States defines an HW index that combines temperature and relative humidity, issuing a heat alert when the HW index exceeds 40.5°C for at least 3 hours in two consecutive days during the daytime, or when it is anticipated to exceed 46.5°C at any time. The Royal Netherlands Meteorological Institute stipulates that an HW event occurs when the daily maximum temperature is above 25°C for more than five consecutive days, with at least three of those days having a maximum temperature exceeding 30°C. In contrast, the China Meteorological Administration (CMA) defines an HW event as a period when the daily maximum temperature exceeds 35°C for three consecutive days. In this study, the HW criteria published by the CMA were finally adopted. Considering that the daily maximum temperature at

urban stations can be influenced by urbanization, this study utilizes the daily maximum temperature from reference stations to identify HW events. During the summer, if more than two reference stations experience an HW event on a given day, the day during the HW event is defined as an HW day; otherwise, it is considered an NHW day."

In addition, the threshold analysis you emphasized, based on long-term climate data during HW periods, indeed represents a significant scientific issue. This area of research has also sparked further explorations, such as the use of multivariate approaches to identify HW events (Kuglitsch et al., 2010; Chen & Li, 2017; Freychet et al., 2017) and the definition of diurnal/nocturnal compound HW (Nairn & Fawcett, 2013; Wang et al., 2020). We plan to employ different HW definitions in our future work to analyze various types of HW events, which hold significant practical implications and application values for studying the interactions between HW and CUHI, as well as developing effective mitigation strategies.

Finally, regarding the identification of heatwaves, you have made a valid point that we overlooked the potential influence of urbanization on daily maximum temperatures recorded at urban stations. We sincerely apologize for this oversight. Following your advice, we have re-examined the HW events using temperature series from reference stations that were less affected by urbanization. The updated findings revealed that the number and duration of HW events in 2019 were overestimated to some extent. We have accordingly revised the relevant figures and statistics in our revised manuscript.


*Response:* Thank you for your insightful comments. According to your suggestions, I have thoroughly revised the captions and legends of all figures.

Taking Fig. 11 as an example, I have made the following improvements:

Fig. 11 Caption: I have clarified the meaning of the vertical short lines on the X-axis, ensuring that readers understand their significance and purpose.

Fig. 11c Legend: I have added a clear explanation of interpolated colors. This should help readers interpret the data presented in this figure more accurately.

In addition, I have added a color legend directly within Fig. 11c.

I believe that these revisions will significantly improve the readability and comprehensiveness of our manuscript. Once again, thank you for your constructive feedback.

**7. It seems to me that it is difficult to conclude from these figures about the influence of mountain or valley breezes on the CUHII knowing that the phenomenon of CUHI has a marked diurnal cycle. On the other hand, the effect of the wind on the UHI may be delayed over time: if there was wind during the day, there is less heat accumulation and then possibly less UHI at night. Same for urban parameters: it's very interesting to see how the different urban parameters rank in terms of their influence on CUHII. However, I wonder about the relevance of comparing this for the"mountain breeze" and "valley breeze" cases. The CUHI phenomenon is different during the day and at night, and is not related to the same physical processes, so it seems difficult to draw relevant**

**conclusions from these comparisons.**

*__Response:__* I apologize for any lack of clarity in my previous response. I would like to clarify further the relationship between the mountain-valley breeze and the ΔCUHII using the examples of four representative stations and you can find the details below.

As shown in Fig. 12 and 13, during the mountain breeze phase (from 05:00 to 10:00), we observed high ΔCUHII values when wind speeds were relatively low. Conversely, during the valley breeze phase (from 11:00 to 10:00 the next day), the ΔCUHII values decreased when wind speeds increased significantly. This observation strongly suggested that wind speed was a factor that influenced the ΔCUHII. To further validate this, we examined the comparison between S1 and S2, two stations located in similar urban north. During the mountain breeze phase, S1 experienced higher wind speeds than S2, and correspondingly, S1 exhibited lower ΔCUHII values than S2. This finding aligned with our hypothesis that wind speed played a pivotal role in modulating the ΔCUHII (Yang et al., 2023; Xue et al., 2023). Moreover, we conducted a detailed analysis by plotting scatter diagrams of the wind speed versus the ΔCUHII from 2016 to 2020, as presented in Fig. R4. This analysis revealed a clear negative correlation between the two variables, confirming that as wind speed increased, the ΔCUHII tended to decrease. This observation reinforced the idea that wind speed was a key factor influencing the ΔCUHII. Additionally, the wind direction might also play a role in the thermal environment (Xie et al., 2022). For instance, during the mountain breeze phase, when the wind blew from urban north to urban south, S3, located in the urban south, experienced a stronger ΔCUHII compared to S1 in the urban north. In summary, the ΔCUHII might be influenced by both the speed and direction of the mountain-valley breeze.

We fully acknowledge that the diurnal cycle of CUHII, as you mentioned, is a confounding factor. Indeed, disentangling the effects of the diurnal cycle from those of the mountain-valley breeze represents a substantial challenge. In the future, we plan to utilize the Weather Research and Forecasting (WRF) model to conduct sensitivity tests with varying wind speeds, with the aim of isolating and investigating the independent influence of wind speed on the ΔCUHII.

[Figure]

**Figure: R4 Scatter plots of the wind speed and the ΔCUHII at urban stations from 2016 to 2020 during the mountain breeze phase and the valley breeze phase.**

In addition, thank you very much for acknowledging my work on ranking the impact of various urban parameters on the ΔCUHII. I sincerely apologize for any lack of clarity in my previous presentation.

As you pointed out, the ΔCUHII curves during the mountain breeze phase and valley breeze phase are indeed markedly distinct. Our intention in comparing the contribution rankings of urban morphology during these two phases was to delve deeper into the driving mechanisms of urban morphology on the synergistic effects. To address your concern and enhance clarity, we have specifically created a schematic diagram that illustrates how the mountain-valley breeze and urban morphology modulate the ΔCUHII in Beijing (as shown in Fig. R5).

[revised manuscript text omitted]

**Minor comments:**

**P3, L69**

**"... more than 1,400 km2 ... "**

*Response:* According to your comments, "... more than 1,400 km2 ... " was revised as "... more than 1,400 km$^2$ ... ".

I have carefully addressed each of your minor comments and double-checked the entire manuscript for any other potential issues.

**P3, L71**

**"The altitudes of those mountains exceede 2,000 meters. "**

*Response:* "... exceede ... " was revised as "... exceed ... ".

**P3, L71**

**"The northeastern part comprises ... "**

*Response:* "... part ... " was revised as "... region ... ".

**P3, L73**

**Please clarify what you mean by "weak weather system"**

*Response:* Clear information about "weak weather system" has been added in the revised manuscript.

The weak weather system here mainly refers to a weather system with no or few clouds (You et al., 2006; Liu et al., 2009; Dong et al., 2017) and is characterized by clear weather and strong solar radiation.


**P5, L90**

**"... within the research buffer areas of the target stations" This information should be introduced later in the text once the stations have been presented.**

*Response:* I apologize for the misplacement of the phrase "the target stations" in the text. Your suggestion is spot on, and I have revised the text accordingly.

**P5, L94-95**

**Replace "encompasses" (not really appropriate) and clarify what you mean by "related elements"**

**Response:** We replaced "encompasses" with "includes", and the "related elements"

referred to specific meteorological factors such as humidity and precipitation.

**P5, L110**

**"... otherwise, it was considered as a non heat wave (NHW) day. " Are NHW days defined for the whole year or just for the summer period?**

**Response:** To clarify, the NHW days in this context are indeed defined specifically for the summer period in the revised manuscript.

**P5, L111-112**

**"... by selecting reference stations for ground temperature observations and urban stations" Please clarify this sentence, do you mean "... by selecting urban reference stations for retrieving near-surface air temperature observations"?**

*Response:* According to your comment, I have revised the relevant section in the revised manuscript as below.

"In general, scholars define CUHII as the temperature difference between the urban station and the reference station."

**P5, L113**

**"... located outside of a 50km radius" Please add a space between "50" and "km" (and do the same everywhere in the rest of the text)**

*Response:* Amended and thanks.

**P5, L112-114**

**Based on which temperature distribution (day, night, average)? Why not base it on the land use map?**

*Response:* We apologize for any confusion caused by our previous expression. Average air temperature and land use were both used as the selection criteria for reference stations, which have been re-described in the revised paper.

**P5, L116**

**"... than the average altitude of the 45 urban stations" How is the station's rural environment defined? From the land use map, I presume ? As I understand it, this means that the other 45 stations (which are not classed as "rural") are all urban, and that they all meet the min distance to the city centre and land use conditions?**

*Response:* Thank you for your valuable suggestion. As you mentioned, the designation of a rural environment is indeed based on land use classification. Additionally, I apologize for any confusion in my previous expression. Regarding the selection of urban stations, we have made supplementary details in the revised manuscript. Line 122-129 in the revised manuscript:

"In general, scholars define CUHII as the temperature difference between the urban station and the reference station (Ren et al., 2007; Shi et al., 2015). The Fifth Ring Road in Beijing, with a length of 98.6 km and an area of approximately 300 km$^2$ (depicted by the blue loop in Fig. 1), essentially covers the primary regions of the built-up area (Yang et al., 2013). Therefore, in this study, we have designated stations within the Fifth Ring Road as urban stations. The selection of reference stations is crucial for calculating the CUHII. In this study, we first identified reference stations with significantly lower temperatures than those of urban stations. Additionally, the reference stations must be located more than 50 km away from the city center, in a rural environment, predominantly situated within areas of sparse trees and shrubs (Yang et al., 2023). They should also be evenly distributed across different directions of the entire city. "


Furthermore, we fully concur with your observation that a clearer description of the evaluation metrics is needed. To address this, we have described the parameters in Table 1 in detail. In addition, we also provide a schematic diagram of key indicators (Fig. R6) to further improve clarity.

[Figure]

**Fig. R6 The schematic diagrams and mathematical formulas of spatial morphological indicators.**


**The presentation and analysis of Fig 2 are extremely confusing. Personally, I don't understand what is presented here. Is it a difference between urban and rural areas, given that the aim is to study "urban warming excess" ?**

*Response:* Thank you for your detailed feedback on our paper. I have revised the figure caption and legends to provide a more straightforward and accurate description. Fig. 2a now clearly states that it represents the temperature difference between urban stations and reference stations, which we refer to as the CUHII. This metric is a crucial indicator of urban warming excess, as it measures the excess warming in urban

areas compared to their surrounding rural or natural environments.

Fig. 2b and Fig. 2c show the number and duration of HW events, respectively, based on the temperatures recorded at the reference stations. These metrics represent the climate background of the study region.

**P9, L180-189 and Fig. 3**

**• CUHII values are relatively low for a city like Beijing. We would expect higher intensities, particularly during heatwaves. How do you explain this? And could it have something to do with the methodology?**

*Response:* Thank you for your insightful comments and feedback on our manuscript.

In our analysis, we observed that the maximum CUHII during HW periods was 2.06°C, compared to 1.32°C during NHW periods. There is a significant increase of 59.33% in the average daily CUHII during HW periods compared to NHW periods. The maximum ΔCUHII reached 0.76°C. These findings are generally consistent with previous studies conducted by Jiang et al. (2019) and Zong et al. (2020).

In response to your question about the ΔCUHII values during HW periods being relatively low for a city like Beijing, we have specifically reviewed relevant research on the ΔCUHII during HW periods in other major cities worldwide in Tab. R1. During HW periods from 2013 to 2018, Shanghai experienced an increase of approximately 0.9°C in CUHII. Beijing and Guangzhou recorded increases of 0.9°C and 0.8°C, respectively (Jiang et al., 2019). In Guangzhou, China, the average CUHII increased by 103% (around 0.9°C) during HW periods compared to NHW periods (Luo et al., 2023). Lanzhou experienced a 103.6% increase in average CUHII (about 1.22°C) (Xue et al., 2023). Athens, Greece, had an average increase of 3.5°C, with peaks reaching 8.0°C under certain conditions (Founda et al., 2017). Seoul, South Korea, witnessed a maximum CUHII increase of 4.5°C (Ngarambe et al., 2020). Across 50 cities in the United States, the average CUHII during HW periods was 0.4~0.6°C higher than during NHW periods (Zhao et al., 2018). During the 2016 HW in the northeastern US, cities like New York, Washington D. C., and Baltimore experienced CUHII amplification of 1.0~2.0°C (Ramamurthy & Bou-Zeid, 2017).

**Tab. R1 The research results on the ΔCUHII in recent years**

| Research city | Country | ΔCUHII | Reference |
|---|---|---|---|
| Guangzhou | China | 0.8–0.9°C | Jiang et al., 2019; Luo et al., 2023 |
| Beijing | China | 0.9°C | Jiang et al., 2019 |
| Shanghai | China | 0.9–1.26°C | Ao et al., 2019; Jiang et al., 2019; Yang et al., 2023 |
| Lanzhou | China | 1.2°C | Xue et al., 2023 |
| Athens | Greece | 3.5°C–8.0°C | Founda et al., 2015 |
| 50 cities | United States | 0.4°C–0.6°C | Zhao et al., 2018 |
| Seoul | South Korea | 4.5°C | Ngarambe et al., 2020 |
| New York City, Washington D.C., and Baltimore | United States | 1.0–2.0°C | Ramamurthy & Bou-Zeid, 2017 |

As you've noticed, the ΔCUHII during HW periods in Beijing, as a major city, may not be as pronounced as in some other cities. This can be attributed to various factors beyond urbanization levels, such as climatic background, geographical characteristics, and differing standards for defining HW events, which may contribute to the disparities in research on the synergistic effects of HW and CUHI (An & Zuo, 2021). Indeed, various standards have been employed globally to study HW events. The World Meteorological Organization (WMO) defines an HW event as when the daily maximum temperature exceeds 32°C for more than three consecutive days. The National Oceanic and Atmospheric Administration (NOAA) of the United States, on the other hand, has developed an HW index that incorporates both temperature and relative humidity, issuing heat alerts when this index surpasses 40.5°C for at least 3 hours on two consecutive days during daylight hours, or when it is forecasted to exceed 46.5°C at any time. The Royal Netherlands Meteorological Institute sets a threshold for HW occurrence when the daily maximum temperature remains above 25°C for more than five consecutive days, with at least three of those days exceeding 30°C. Conversely, the China Meteorological Administration (CMA) defines an HW event as a period when the daily maximum temperature exceeds 35°C for three straight days. Moreover, some scholars have explored bivariate approaches to identify HW events, such as methods that consider both daily maximum and minimum

temperatures exceeding specified thresholds over multiple consecutive days (Kuglitsch et al., 2010; Chen & Li, 2017). In light of these diverse definitions, we plan to adopt various HW criteria in our future research to delve deeper into the characteristics of HW activity in the Beijing region.

Thank you again for your valuable comments. We hope these clarifications address your concerns and strengthen our manuscript.


**Section 3.3**

**The city configuration with variability of building densities and heights is interesting.**

**P 16, Fig. 7c**

**According to the figure, D-value (for dense vs open) is stronger for "whole days" case than for both "mountain breeze" and "valley breeze" cases. However, the D-value for "whole days" should be intermediate to the other two cases, if it's calculated as an average over all the hours of the day, right?**

*Response:* We have identified an error where the difference value (dense vs open) for "whole days" inadvertently referenced the difference value for "mountain breeze". We deeply regret any confusion or inconvenience this may have caused.

Rest assured, we have promptly corrected this mistake in Fig 7c. Additionally, during our thorough review of the figures, we also noticed two meaningless blue dots in Fig. 7a and Fig. 7b, which we have removed to ensure the clarity and accuracy of the presentation. We have taken this opportunity to carefully recheck all the figures and tables throughout the manuscript to prevent any further errors.

**P17, L287**

**What do you mean by "during 3D indicators" ? I don't understand this sentence.**

*Response:* I meant to say "among 3D indicators," which I have now corrected in the revised manuscript.

**P17, L287**

**The term "amplified CUHII" could be clearly explained once and then replaced by ΔCUHII (everywhere in the text and figures).**

*Response:* Thank you for your careful review of our manuscript.

As suggested, in the Introduction, we first labeled the acronym ΔCUHII for the first occurrence of amplified CUHII and have consistently used the ΔCUHII throughout the rest of the manuscript to refer to the amplified CUHII. Secondly, we have now provided a comprehensive explanation of the meaning and computation method of the ΔCUHII in section 2.3.1 of the Methods section. Specifically, the ΔCUHII was obtained by subtracting the summer CUHII during the NHW periods from the summer CUHII during the HW periods.

**P18, L289-291**

**You say "Urban morphological indicators had weaker relationships with amplified CUHII during the mountain breeze period but showed stronger correlations with amplified CUHII during the valley breeze period. "**

**According to Fig. 8, the effect of urban indicators on ΔCUHII according to the breeze circulations is the opposite of what is written here: the effect is stronger during mountain breezes.**

*Response:* Thank you for taking the time to review our manuscript and for bringing this oversight to our attention. You are absolutely correct in pointing out that we inadvertently reversed the description of the relationships between the urban morphological indicators and the ΔCUHII during the mountain breeze phase and the valley breeze phase.

We apologize for this mistake. The revised sentence is shown below:

"During the mountain breeze phase, the relationship between the urban morphological indicators and the ΔCUHII was stronger, whereas during the valley breeze phase, this relationship was weaker."

**Fig. 9**

**We don't understand what is presented here, the names of the axes are not explicit and the legend is not detailed enough. I presume it is ΔCUHII(OBS) vs ΔCUHII(MODEL) ?**

*Response:* Thank you for your thorough review of our manuscript. You are absolutely correct that the label of the axes in Fig. 9 has been inadequate, causing confusion.

The vertical axis represents the observed change in a parameter denoted as the ΔCUHII (OBS), while the horizontal axis depicts the corresponding modeled change, labeled as the ΔCUHII (MODEL). We apologize for the oversight. To rectify this, we have revised the axis labels and legends of Fig. 9.

**P18, 297-303**

**• Linear model: It's rather debatable to say here that the linear model is good (especially as you go on to say that it doesn't perform well...). The RMSE of 0.14°C is rather high given the average values. You should adapt your comments**

*Response:* Thank you for your careful review of our manuscript. You are absolutely correct that our previous statement was not accurate, and we sincerely apologize for any confusion or misinterpretation it may have caused.

We have now revised the manuscript accordingly, specifically in line 342 to 345 as below:

"As depicted in Fig. 9a, the linear model yielded a coefficient of determination ($R^2$) of 0.44 and a root mean square error (RMSE) of 0.14°C, indicating a relatively large modeling error. Consequently, while the linear model provided a foundational framework for modeling the ΔCUHII, it might not be the most optimal choice for our study. "

**• SVR and RF: both models overestimate high values and underestimate low values, why?**

*Response:* Thank you for your insightful questions regarding the modeling biases in our study. We have added two error bars (30%) in the modeling results for SVF and RF to facilitate observation. As you have pointed out, we indeed observe a tendency of overestimation at higher values and underestimation at lower values for both SVF and RF. Notably, the underestimation is more pronounced within the temperature range of 0.0~0.2°C.

[Figure]

**Fig. R5 The performance of the SVR model (a) and RF model (b) in predicting the ΔCUHII.**

We believe that two primary factors may contribute to this phenomenon. Firstly, dataset imbalance is a critical factor that could explain the observed biases. According to previous studies (He & Garcia, 2009), when the data set contains more samples from one category (high-value) than other categories (low-value), it often affects the prediction results of the model. Moreover, if the target variable's distribution is nonlinear, with distinct characteristics between high-value and low-value regions, models may fail to adequately capture these complexities, further exacerbating the modeling biases (Breiman, 2001).

Secondly, the specific characteristics of our dataset, particularly the autocorrelation among the urban morphology parameters used as predictors, could also play a role. Autocorrelation among predictor variables can introduce complexities that complicate the modeling process (Dormann et al., 2013). In conclusion, we believe that the observed prediction biases are likely due to a combination of dataset imbalance and

the specific characteristics of our dataset.

It is intriguing to note that, the RF model exhibited superior modeling performance for the ΔCUHII compared to the SVF model when applied to the same dataset. We recognize that further investigation is needed to fully understand and mitigate the modeling biases in our study. Your comments have been invaluable in guiding our analysis, and we are committed to incorporating your suggestions into our future research efforts.


**P20, L331**

**"... the importance of SVF and BCR in the 2D and 3D indicators ... " should be revised as "... the importance of SVF and BCR in the 3D and 2D indicators ... "**

*Response:* Corrected.

**Fig. 11**

**• Again we don't understand what is presented here, what is the partial dependence? clarify what is presented both in the text and in the caption. Is it ΔCUHII on y-axis?**

*Response:* I apologize for the lack of clarity in our previous submission. I have made the necessary clarifications in both the text and figure captions.

Partial dependence, in the context of machine learning, refers to the assessment of the relationship between a single feature and the model's predicted outcome, while all other features in the dataset are held constant (Friedman, 2001). This function represents the effect of selected explanatory variables and can be used to interpret

"black box" models (Cutler et al., 2007; Shiroyama & Yoshimura, 2016). In Fig. 11a-11b, the x-axis represents various feature values, and the y-axis represents the partial dependence value.

• **The range of y-axis is different in the different plots, you should use the same.**

*Response:* Corrected.

• **What do the small vertical lines on the x-axis represent?**

*Response:* I apologize for the lack of clarity regarding the small vertical lines on the x-axis in Fig. 11a-11b. I have supplemented a detailed explanation in the figure caption. In machine learning, the small vertical lines along the x-axis, known as rug plots, represent the distribution of the feature values. They provide a visual representation of the data's density and can help identify regions of high or low data concentration.

• **For (c) panel, do you think it is relevant to interpolate the data? I would use symbols instead. Also you should add a color legend.**

*Response:* Thank you for your insightful comments. Based on your suggestion, I have added color legends in Fig. 11c. All discussions in Fig. 11a-11b have centered around the impact of individual variables on the predicted values, without considering how these variables might simultaneously influence the prediction. Therefore, in Fig11. c, we have introduced the two-way PDP (Partial Dependence Plot), which allowed us to consider the range of values for two feature variables simultaneously and demonstrates their joint influence on the prediction of the dependent variable.

[Figure]

**Figure: 11 (a-b) Partial dependence plots of the ΔCUHII on BCR and SVF. The red line represents the fitted curve, while the gray lines indicate the 95% confidence interval. The rug plots (small vertical lines) along the X-axis represent the distribution of the feature values. (c)The two-way plots partial dependence of the ΔCUHII on BCR and SVF. The X-axis represents the BCR feature, while the Y-axis represents the SVF feature. The interpolated colors of the panel, ranging from dark to light, signified the partial dependence decreasing from large to small.**

---

## Author Comment (AC2)

**Response to Reviewer Comments**

Dear Reviewer and Editors:

We are sincerely grateful to the editor and reviewer for their valuable time for reviewing our manuscript. The comments are very helpful and valuable, and we have addressed the issues raised by the reviewer in the revised manuscript. Please find our point-by-point response (in blue text) to the comments (in black text) raised by the reviewer. We have revised the paper according to your comments (highlighted in red text of the revised manuscript).

Sincerely yours,

Dr. Yuanjian Yang, representing all co-authors

**Reviewer #2:This case study work investigated the canopy UHI of Beijing City in the summers of 2016-2020. Based on data from various sources, the authors conducted a systematic analysis, aiming to get a better understanding of the synergy between heat waves and the canopy UHI, as well as the influences of local circulation and urban morphology on this synergy. It is an interesting work with the support of a dense observation network and other fine-resolution data sets.**

**The authors try to consider many different aspects, which in turn requires the combination of many different data sources and analysis methods. This, however, makes the paper just as complex as the climate topic. Without improvement on the structure of this paper, especially the clarity of the data and methods, and the fragmented results presentation, it is difficult to follow the flow of the paper and to grasp the key findings.**

*Response:* Thanks very much for taking time out of your busy days to provide us with such valuable comments that significantly improve the quality of our manuscript. In

line with your comments and suggestions, we have revised our manuscript carefully and prepared a list of point-by-point responses below.

Firstly, the Introduction section is updated carefully to better highlight the central scientific question of this study. In our revised manuscript, we focus on the scientific question of how mountain-valley breeze and urban morphology drive the positive feedback effect between HW and CUHII in the Beijing megacity.

Secondly, we have revised the descriptions of the key concepts and calculation methods for CUHII, HW, mountain-valley breeze, and urban morphology in the Data and methodology section. These updates ensure that readers have a clear understanding of the approach in this manuscript.

Additionally, a summary of the key findings and the corresponding subsequent analysis are appended at the end of each subsection of the section of Results to improve the coherence of this study.

Finally, we have thoroughly revised the sections of Abstract and Conclusion to highlight our main findings. Specifically, we have emphasized the influence of wind speed and direction on the temporal and spatial distribution of the positive feedback effect between HW and CUHII, as well as the driving role of both two-dimensional and three-dimensional urban morphology in the effect.

**Major comments:**

**1. The language needs improvement.**

*Response:* Thank you for taking the time to review my manuscript and provide valuable feedback. I have carefully revised the text to ensure clarity, conciseness, and fluency throughout.

**2. Some details in the data and method section need to be clarified, some decisions need to be justified. I think this paper needs to be better structured, some details in the data and method sections need to be clarified, and the results need to be presented more concisely.**

*Response:* I sincerely apologize for any unclear in the previous version of the

manuscript, particularly in Data and Methodology section.

In response to your valuable suggestions, I have thoroughly enhanced the descriptions in the Data and Methodology section. This includes additional information on the introduction of LCZ data, the definition of HW, the calculation method for CUHII, the computation of the amplified CUHII during HW periods (ΔCUHII), the methodology for calculating mountain-valley breeze, and the explanation of urban morphology indicators. Taking the detailed explanation of the mountain-valley breeze calculation method as an example, the corresponding updated information is shown in lines 154-163 of the revised manuscript:

"Referencing relevant methods (Cao et al., 2015; Zheng et al., 2018), the mountain-valley breeze is extracted and the details are shown as below. Firstly, the hourly wind data from each observation station were decomposed into the components of u (east-west direction) and v (north-south direction). From June to August between 2016 and 2020, the average values of the hourly wind components were calculated, yielding hourly average values $\overline{u}$ and $\overline{v}$. Subsequently, the diurnal average values U and V were obtained by averaging all the hourly average values $\overline{u}$ and $\overline{v}$, respectively. The hourly anomalies u' and v' were then derived by subtracting the diurnal average values U and V from the hourly average values $\overline{u}$ and $\overline{v}$, respectively. The diurnal average values U and V can be interpreted as the systematic wind or background wind, while the hourly average values $\overline{u}$ and $\overline{v}$ can be considered as the actual wind. The local wind u' and v' obtained by subtracting the systematic wind from the actual wind, can be utilized in studies focused on regional local circulations, in particular for the mountain-valley breeze."

Reference:

Cao, J., Liu, X., Li, G., Zou, H.: Analysis of the phenomenon of Lake-land breeze in Poyang Lake area, Plateau Meteorol. Chin., 426–435, 10.7522/J.ISSN.1000-0534.2013.00197, 2015.

Tian, Y., Miao, J.: Overview of Mountain-Valley Breeze Studies in China. Meteorological Science and Technology, 47, 1, 11. https://doi.org/10.19517/j.1671-6345.20170777, 2019.

Zheng, Z., Ren, G., Gao, H. Analysis of the local circulation in Beijing area, Meteorological Monthly, 44, 3, 425–433, https://doi.org/10.7519/j.issn.1000-0526.2018.03.009, 2018b.

**3. Please rephrase the last paragraph of the introduction section to emphasize your key research questions.**

*Response:* I sincerely apologize for the oversight in the final paragraph of the introduction, where the key scientific questions were not adequately described. Your observation is absolutely right, and I appreciate your guidance in improving the clarity of our work.

To address this, we have expanded the introduction section to provide a more detailed overview of the existing research in this field. We have specifically highlighted that previous studies predominantly centered on the spatiotemporal variations of the $\Delta$CUHII in the Beijing megacity, leaving a gap in knowledge regarding the driving mechanisms of local circulation and urban morphology on amplifying CUHII during HW periods.

Recognizing this gap, we have identified the exploration of this very topic as the scientific question of our study. Our research aims to investigate how mountain-valley breeze and urban morphological characteristics drive the significant feedback effect between HW and CUHI in Beijing's urban environment.

**4. Please present your results more concisely while sticking to your key research questions. Try not to overwhelm readers with less important details in numbers, instead, try to summarize and convey the key results, to better connect different fragments of result presentations so that the storyline flows better.**

*Response:* Thanks very much for your valuable comment. We have carefully considered your feedback and made several improvements to address your concerns.

Firstly, we have removed the unnecessary numerical details as you requested to simplify the text.

Secondly, we have expanded the Introduction section to more prominently outline the

key research questions.

In addition, a summary of the key findings and the corresponding subsequent analysis are appended at the end of each subsection of the section of Results to improve the coherence of this study.

Finally, we have refined the Conclusion section to ensure it clearly and concisely summarizes our main findings.

We are grateful for your attentive review and invaluable suggestions, which have significantly contributed to enhancing the quality of our manuscript.

**Minor comments:**

**P1Line21: "insight to understand the driving mechanisms…" maybe "insight into understanding the driving mechanisms." or "insight into the driving mechanisms"**

_**Response:**_ We greatly appreciate the time and patience you have taken to provide your insights on our study. I deeply apologize for the small errors and oversights that you have pointed out. I have carefully addressed each of your comments and double-checked the entire manuscript for any other potential issues.

**P1Line16: "On a street scale" , maybe "on the street level"**

_**Response:**_ Corrected.

**P1Line25: "hasbecome" space in between missing**

_**Response:**_ Corrected.

**P2Line43: "Few studies…" Do you mean "A few studies…" or do you want to say this has not been studied.**

_**Response:**_ Thanks very much for your valuable comment. What I intended to convey was that a limited number of scholars have conducted relevant studies on this topic. "Few studies" was revised to "a limited number of scholars".

**P2Line36:** "The rate of contribution of urbanization to the excessive mortality caused by high temperatures can reach more than 45% in the high-density urban areas"

-> The contribution rate of urbanization to excessive mortality caused by high temperatures can exceed 45% in high-density urban areas.

*Response:* Thank you for your patient guidance. The revisions have been made accordingly.

**P2Line47:** "Overall, the current understanding of the mechanisms through which local circulations modulated the amplified CUHII during HW periods is still in the exploratory stage."

-> Overall, the current understanding of how local circulations modulate the amplified CUHII during heatwave periods is still in the exploratory stage.

*Response:* Corrected.

**P2Line49:** "impact" not necessary

*Response:* Corrected.

**P2Line59:** "However, LCZs are a comprehensive indicator of urban morphology, and the aforementioned studies have not quantified the contribution of different urban morphological parameters to the local thermal environment, nor have they taken into account the nonlinear driving effects of 60 urban morphology on the local thermal environment (Alonso & Renard, 2020; Chen et al., 2022)."

-> However, while LCZs are a comprehensive indicator of urban morphology, the aforementioned studies have neither quantified the contributions of different urban morphological parameters to the local thermal environment nor considered the nonlinear driving effects of urban morphology on the local thermal environment.

*Response:* Thank you for your valuable comment. The manuscript has been revised carefully according to your comment.

**P2Line63: "Currently, it is still matter of debate the roles of local circulations and urban morphology in amplifying CUHI in megacities during HW periods."**
**-> Currently, the roles of local circulations and urban morphology in amplifying CUHI in megacities during heatwave periods are still a matter of debate**
*Response:* Corrected.

**P2Line64: "The main objective of this study is considering as case study the megacity of Beijing, using high-density automatic weather stations (AWS) observations."**
**-> The main objective of this study is to use high-density automatic weather station (AWS) observations to analyze the megacity of Beijing as a case study.**
*Response:* Corrected.

**P3Line70: "The terrain of Beijing is exceptionally complex, northly bounded by Yan Mountains by Taihang Mountains in the west."**
**-> The terrain of Beijing is exceptionally complex, bounded to the north by the Yan Mountains and to the west by the Taihang Mountains.**
*Response:* Thank you for your patience. The manuscript has been revised.

**P4: FIG1c, not built-up -> Non-built-up**
*Response:* Corrected.

[Figure]

**Figure 1: Overview of study area. (a) Terrain and land use of Beijing. (b) Distribution of urban stations and reference stations in the built-up area of Beijing. (c) Empirical examples of the typical LCZ types.**

**P5Line85:** "released by Professor Yang and Professor Huang of Wuhan University"This is not necessary, you already cited the corresponding publication.

**_Response:_** I apologize for any confusion caused by the previous expression. I have made the appropriate revision in the revised manuscript as below:

"The annual China Land Cover Dataset (CLCD) is a dynamic data set accounting for land use in China released by Yang & Huang (2021)."

**P5Line85:** "made" maybe "produced"

**_Response:_** Corrected.

**P5Line89: "The building skyline and floor data of the electronic map were extracted using Python language." Electronic map from which provider?**

*Response:* Thanks very much for your valuable comment. The electronic map data used in our study was sourced from Gaode Maps. I have added the information in the revised manuscript.

**P5Line90: "was estimated to be 3 m"**

**was set to be 3 m**

*Response:* Corrected.

**P5Line112: "This study identified stations that were less influenced by the urban effect" Selecting reference stations is always a challenge and the criteria are subject to the decision of different choices that might be biased. From the map on Fig1, it seems these reference stations are rather close to built-up areas.**

*Response:* Thanks very much for your valuable comment. In our revised manuscript, the details of the method for identifying urban stations and reference stations are added in the section of Data and methodology.

In general, scholars define CUHII as the temperature difference between the urban station and the reference station (Ren et al., 2007; Shi et al., 2015). Thus, identifying the urban stations and rural reference stations is very important for investigating urban climate. In the region of our study, Beijing has undergone massive and rapid urbanization, with its urban space continually expanding into the suburbs over the past few decades. Currently, Beijing boasts a population of 20 million and a built-up area spanning 1400 km². Due to this expansion, a swift transportation system has become imperative for urban development, prompting Beijing to commence the construction of a Multiple-ring-road system since the 1990s (Wang et al., 2010). These rings effectively represent the radial expansion of urban zones, with varying population and building densities. Notably, the Fifth Ring Road, with a length of 98.6 km and a built-up area of approximately 300 km² (depicted as the blue ring in Fig. R1), encompasses the primary regions of the built-up area (Yang et al., 2013). The

distribution characteristics of average air temperature around the built-up area of Beijing are illustrated in Fig. R1, showing that the high-temperature zone in the city center aligns closely with the extent of the Fifth Ring Road. Additionally, the proportion of densely built-up areas within the Fifth Ring Road exceeds 85%, significantly higher than that outside this ring. Therefore, we have designated stations within the Fifth Ring Road as urban stations in this study.

The identification of reference stations is shown below. Firstly, the reference stations should have significantly lower temperatures than those of urban stations, based on the spatial distribution characteristics of average temperatures. Secondly, the reference stations must be located more than 50 km away from the city center, in a rural environment, predominantly situated within areas of sparse trees and shrubs (Yang et al., 2023). Thirdly, the reference stations should also be evenly distributed across different directions of the entire city. According to these criteria, eight reference stations were selected (green plot in Fig. R2), with an average altitude of 39.6 m, which is only 8.8 m lower than the average altitude of 45 urban stations (red plot in Fig. R2).

[Figure]

**Fig. R1 The distribution characteristics of average air temperature around the built-up area of Beijing.**

[Figure]

**Fig. R2 Spatial distribution of urban and reference stations in Beijing.**

**Reference:**

Yang, P., Ren, G., Liu, W.: Spatial and temporal characteristics of Beijing urban heat island intensity. Journal of Applied Meteorology and Climatology, 52, 8, 1803-1816, http://doi.org/10.1175/JAMC-D-12-0125.1, 2013.

Ren, G., Chu, Z., Chen, Z., Ren, Y.: Implications of temporal change in urban heat island intensity observed at Beijing and Wuhan stations, Geophysical Research Letters, 34, 5, https://doi.org/10.1029/2006GL027927, 2007.

Shi, T., Huang, Y., Shi, C., & Yang, Y.: Influence of Urbanization on the Thermal Environment of Meteorological Stations: Satellite-observational Evidence, Advances in Climate Change Research, 1, 7–15, https://doi.org/10.1016/j.accre.2015.07.001, 2015.

Yang, Y., Guo, M., Wang, L., Zong, L., Liu, D., Zhang, W., Wang, M., Wan, B., Guo, Y.: Unevenly spatiotemporal distribution of urban excess warming in coastal Shanghai megacity, China: Roles of geophysical environment, ventilation and sea breeze, Building and Environment, 235, https://doi.org/10.1016/j.buildenv.2023.110180, 2023.

Wang, X., Li, X., Feng, Z.: Research on Beijing urban expansion based on the principle of information entropy. China Population,Resources and Environment, S1, 88–92, https://doi.org/CNKI:SUN:ZGRZ.0.2010-S1-024, 2010.

**P6Line124: "Calculation of mountain-valley breeze", I suggest adding more technical details on this in the supplementary information.**

*Response:* Thank you for your insightful comments. I have now included additional technical details on this topic in the line 153-162 of the revised manuscript as below:

"Referencing relevant methods (Cao et al., 2015; Zheng et al., 2018), the mountain-valley breeze is extracted and the details are shown below. Firstly, the hourly wind data from each observation station were decomposed into the components of u (east-west direction) and v (north-south direction). From June to August between 2016 and 2020, the average values of the hourly wind components were calculated, yielding hourly average values $\bar{u}$ and $\bar{v}$. Subsequently, the diurnal average values U and V were obtained by averaging all the hourly average values $\bar{u}$ and $\bar{v}$, respectively. The hourly anomalies u' and v' were then derived by subtracting the diurnal average values U and V from the hourly average values $\bar{u}$ and $\bar{v}$, respectively. The diurnal average values U and V can be interpreted as the systematic wind or background wind, while the hourly average values $\bar{u}$ and $\bar{v}$ can be considered as the actual wind. The local wind u' and v' obtained by subtracting the systematic wind from the actual wind, can be utilized in studies focused on regional local circulations, in particular for the mountain-valley breeze."

**Reference:**

Cao, J., Liu, X., Li, G., Zou, H.: Analysis of the phenomenon of Lake-land breeze in Poyang Lake area, Plateau Meteorol. Chin., 426–435, 10.7522/J.ISSN.1000-0534.2013.00197, 2015.

Zheng, Z., Ren, G., Gao, H. Analysis of the local circulation in Beijing area, Meteorological Monthly, 44, 3, 425–433, https://doi.org/10.7519/j.issn.1000-0526.2018.03.009, 2018b.

**P6: Table1, "Building cover ratio, which represents the proportion of the roof of the building to that of the entire study area." This is rather confusing, what do you mean by "the entire study area"? What do you mean by "study area", do you mean the entire Beijing urban are or only the area within the buffer zone?**

*__Response:__* I apologize for any confusion caused by my previous phrasing in Table 1. To address your concerns, I have replaced all occurrences of "study area" and "entire study area" with "buffer zone" in Table 1. In addition, I have explicitly stated in the text that the buffer zone refers to a 500 m area surrounding the AWS (Oke, 2004). This clarification should help readers understand the scope and context of our analysis.

**Reference:**

Oke, T. R.: Initial guidance to obtain representative meteorological observations at urban sites. University of British Columbia, Vancouver, 2004.

**P6Line131: "to measure the morphological characteristics around AWS", up to which distance surrounding the AWSs?**

*__Response:__* Thank you for your thoughtful comments on our manuscript. We have supplemented the information in line 166 of the revised manuscript as below:

"to measure the morphological characteristics of buildings within a 500 m buffer zone surrounding the AWS"

**P6: Table1,"Number of patches." What patch?**

*__Response:__* Thank you for bringing this to our attention.. To clarify and improve the clarity of our manuscript, we have revised the term to "Number of building patches."

**P6: "Table 1: Summary of the spatial morphological parameters." I know it is rather complicated and might require too much effort, but it would be much better to give the equations for calculating these factors, especially since you failed to describe them clearly in the text.**

*__Response:__* Thank you for your patience in reviewing our manuscript. We completely agree with your assessment regarding the need for clearer descriptions of some of the indicators in Tab. 1. To address this, we have described the parameters in Table 1 in detail. In addition, we also provide a schematic diagram of key indicators (Fig. R3) to further improve clarity.

**Table 1: The 2D and 3D spatial morphology indicators involved in this paper.**

| Indicators | Description |
| --- | --- |
| **2D** | |
| BCR | Building cover ratio represents the proportion of the roof of the buildings to that of the buffer zone. |
| NEAR | Mean distance between adjacent buildings. A lower value of this metric indicates a higher density of buildings. |
| NP | Number of buildings patches indicates the degree of fragmentation of buildings within a given area. |
| SPLIT | Splitting index represents the degree of separation of landscape segmentation. The greater the value, the more fragmented the landscape. |
| AI | Aggregation index, which represents the connectivity between patches of each type of landscape. The smaller the value, the more discrete the landscape. |
| L/W | Length-width ratio of buildings is a metric that represents the shape characteristics of buildings. |
| **3D** | |
| H | The height of buildings represents the average height of all buildings in the buffer zone. |
| H-max | Maximum height of buildings in the buffer zone. |
| H-std | The standard deviation of building height in the buffer zone. |
| FAR | Floor area ratio represents the ratio of the sum of gross floor area to total buffer zone. The higher the FAR, the greater the amount of building floor area per unit of land area. |
| CI | Cubic index represents the ratio of the building volume to the total study volume. It indicates a higher degree of built-up density |

| | or spatial occupation within the buffer zone when the value is larger. |
|---|---|
| SVF | Sky view factor represents the ratio of radiation received by a planar surface from the sky to that received from the entire hemispheric radiating environment. It ranges from 0 to 1, with 0 indicating complete obstruction and 1 indicating complete exposure. |

[Figure]

**Figure: R3 The schematic diagrams and mathematical formulas of spatial morphological indicators.**

**P7: Table1, "in the buffer zone", you should really mention the buffer zone at the beginning of section 2.3.3, and as per my comment above, what is the size of the buffer zone? How do you define these buffer zones? This needs to be mentioned in the text.**

***Response:*** Thank you for your careful review and constructive feedback. I have taken your comments seriously and have revised the manuscript accordingly as below:

"Here, we selected six indicators of horizontal morphology and six indicators of vertical morphology to measure the morphological characteristics of buildings within a 500 m buffer zone surrounding the AWS (Oke, 2004)"

**Reference:**

Oke, T. R.: Initial guidance to obtain representative meteorological observations at urban sites. University of British Columbia, Vancouver, 2004.

**P7Line151: "The impact of urban spatial morphology on urbanization bias" not sure what you mean by "urbanization bias"**

*Response:* The "urbanization bias" was corrected to "the ΔCUHII".

**P8Line165: "relatively weaker urban excess warming," Not sure if the HW should be the basis for defining urban excess warming. My intuition would be that CUHII is a measure of urban excess heat/warming.**

*Response:* I would like to extend my sincere apologies for the inappropriate phrasing in my manuscript. The relevant content was revised in line 201-204 in the revised manuscript as below:

"The most prominent years for urban excess warming, specifically in terms of CUHII, were 2016 and 2019, with intensities of 1.00°C and 0.97°C respectively. In these two years, the HW numbers were 3 times and 2 times, while the HW duration were 9 days and 10 days respectively."

**P8: Fig2, it seems there is no correlation between CUHI and HWs. How is the CUHII calculated? By averaging all the hourly CUHII within one month?**

*Response:* Thank you for bringing this clarification to our attention. The method used to calculate CUHII was specifically based on comparing the air temperature differences between urban stations and reference stations during the summertime.

$$CUHII = T_{urban} - T_{reference} \qquad (1)$$

CUHII is the canopy urban heat island intensity during the summertime, $T_{urban}$ is the air temperature of the urban stations during the summertime, and $T_{reference}$ is the summer air temperature of the reference stations during the summertime. We have calculated the diurnal variation of the temperature difference between the urban station and the reference station during the summertime (i.e., the diurnal variation of

CUHII). In Fig. R4, the blue line represents the diurnal variation of summer temperature at the urban station, while the green line depicts the diurnal variation of the temperature at the reference station. By calculating the difference between these two stations, we obtained the diurnal variation of CUHII during the summertime.

[Figure]

**Fig. R4 The diurnal variation of CUHII in the built-up areas of Beijing during the summertime.**

**P12Line205: "the mountain-valley breeze strongly impacts the thermal dynamic field near-surface of Beijing megacity"**

**-> the mountain-valley breeze strongly impacts the near-surface thermal dynamic field of the Beijing megacity**

_**Response:**_ Corrected.

**P14: Fig6d, the wind speed in UN is still larger than in US, what are the possible reasons and implications?**

_**Response:**_ Thank you for your insightful comments. Indeed, as evidenced by previous studies (Dou et al., 2014), showcasing higher wind speeds in UN compared to US during the summer valley breeze phase, as shown in Fig. R5.

We speculate that this phenomenon is intimately tied to the mechanics of the mountain-valley breeze. As illustrated in Fig. R6, during the valley breeze phase, the hillside, exposed to abundant solar radiation, experiences a significant temperature rise, acting as a relative heat source. In contrast, the air above the urban, located further from the ground, warms less, functioning as a relatively cold source. This

thermal contrast between the hillside and the city triggers a thermal circulation, with warm air rising from the hillside and flowing over to the city's upper layers, while cooler air from the city ascends the hillside to replace it.

Notably, the larger thermal gradient between UN and the hillside, compared to that between US and the hillside, could explain the observed higher wind speeds in UN during the valley breeze phase. However, this is merely a hypothesis at this stage.

To further explore the diurnal characteristics of wind fields in the built-up area of Beijing, we plan to conduct sensitivity tests using the WRF model. Specifically, we aim to investigate how the wind patterns in the UN and US would differ if the mountainous terrain were removed from the simulation.

Thank you again for your valuable feedback, which has enriched our understanding of this complex phenomenon.

[Figure]

**Fig. R5 Wind fields of mountain-valley breeze in summertime of the Beijing megacity (Dou et al., 2014).**

[Figure]

**Fig. R6 Schematic diagram of the differences in wind fields between the UN and US during the valley breeze phase (self drawn).**

**Reference:**

Cai, X., Guo, Y., Liu, H., Chen, J.: Flow Patterns of Lower Atmosphere over Beijing Area, Acta Scientiarum Naturalium Universitatis Pekinensis, 38, 5, 698–704, https://doi.org/10.3321/j.issn:0479-8023.2002.03.015, 2002.

Dong, Q., Zhao, P., Wang, Y., Miao, S., Gao, J.: Impact of Mountain-Valley Wind Circulation on Typical Cases of Air Pollution in Beijing. Environmental Science, 38, 6, 2218–2230, https://doi.org/10.13227/j.hjkx.201609231, 2017.

Dou, J., Wang, Y., Miao, S.: Fine Spatial and Temporal Characteristics of Humidity and Wind in Beijing Urban Area. Journal of Applied Meteorological Science, 25, 5, 559–569, https://doi.org/10.11898/1001-7313.20140505, 2014.

**P10Line185: "It should be noted that the ΔCUHII remained positive throughout the daytime and nighttime," It is not always true, for example in 2018 in Fig3c.**

*Response:* Thanks very much for your valuable comment. I have revised my language in the line 219-228.

"CUHII started to slowly decrease from 06:00 Beijing Time (BJT) and hit its lowest point at 16:00 BJT. Then, CUHII gradually increased and remained at a high plateau consistently from 22:00 until 05:00 the next day. The diurnal variation of the ΔCUHII was also examined in this study. Apart from 19:00 in 2016 (Fig. 3a) and 2018 (Fig. 3c), the hourly ΔCUHII values for all other years were positive. Taking the annual average as an example (Fig. 3f), the CUHII ranged between 0.18 and 2.06°C during HW periods, which is much larger than that during NHW periods varied between 0.03 and 1.32°C. In particular, the average daily CUHII during HW periods exhibited a significant increase of 59.33% compared to that during NHW periods. The maximum ΔCUHII was 0.76°C, occurring at 00:00 BJT, while the minimum ΔCUHII was 0.05°C, observed at 19:00 BJT. It should be noted that the ΔCUHII remained positive throughout the daytime and nighttime, indicating the persistent synergies between HW and CUHI in the built-up area of Beijing."

**P15Line255: "Therefore, on an urban scale, the turning mountain valley breeze caused horizontal transport of heat inner city, resulting in the north-south asymmetrical pattern of urban excess warming during HW periods." do you mean "horizontal heat transport of heat of the inner city", or just "horizontal heat transport in the city"?**

*Response:* I apologize for the unclear in my previous phrasing. What I mean is "horizontal heat transport in the built-up area". I have revised the text in the revised manuscript.

**P16: Fig7c, "Difference value (D-value) in CUHII across different urban configuration structures." It reads a bit weird.**

*Response:* Thank you for bringing this to my attention. I apologize for the confusion

caused by the previous phrasing in Fig. 7c's caption.

I have revised the caption to provide a clearer description as below:

"Differences in CUHII between compact rise and open rise, and between high rise and low rise."

**P18Line296: "The linear model has shown considerable strength" I suppose a new section should start from here. This is also why I have the impression that the results parts a rather fragmented. You need some connection between them to let the contents flow smoothly so that they form a whole and readers can better follow what you want to convey.**

*Response:* Thank you for your insightful comments.

In response to your suggestion, we have revised the relevant section to provide a more precise and informative description of the linear model's performance. The revised text reads in line 341-343 as below:

"As depicted in Fig. 9a, the linear model yielded a coefficient of determination ($R^2$) of 0.44 and a root mean square error (RMSE) of 0.14°C, indicating a relatively large modeling error. Consequently, while the linear model provided a foundational framework for modeling the $\Delta$CUHII, it might not be the most optimal choice for our study."

Furthermore, we have revised the Results section by appending a summary of the key findings at the conclusion of each subsection and appropriately introducing subsequent analysis to ensure fluency between different sections. Finally, we have refined the Conclusion section to provide a clear and concise summary of all the findings.

**P20Line330: "As the previous text demonstrated"**

**-> As previously demonstrated**

*Response:* Thank you for your patience and guidance throughout the review process. I have carefully rechecked the language in the manuscript once again to ensure clarity and precision.